# A Review of the Recent Developments of Molecular Hybrids Targeting Tubulin Polymerization

**DOI:** 10.3390/ijms23074001

**Published:** 2022-04-04

**Authors:** Oluwakemi Ebenezer, Michael Shapi, Jack A. Tuszynski

**Affiliations:** 1Department of Chemistry, Faculty of Natural Science, Mangosuthu University of Technology, Umlazi 4031, South Africa; re.korede@gmail.com (O.E.); mshapi@mut.ac.za (M.S.); 2Department of Oncology, Cross Cancer Institute, University of Alberta, Edmonton, AB T6G 1Z2, Canada; 3Department of Physics, University of Alberta, Edmonton, AB T6G 2E1, Canada; 4DIMEAS, Politecnico di Torino, 10129 Turin, Italy

**Keywords:** chemotherapy, tubulin, cancer, microtubules, polymerization, inhibition

## Abstract

Microtubules are cylindrical protein polymers formed from *α**β*-tubulin heterodimers in the cytoplasm of eukaryotic cells. Microtubule disturbance may cause cell cycle arrest in the G2/M phase, and anomalous mitotic spindles will form. Microtubules are an important target for cancer drug action because of their critical role in mitosis. Several microtubule-targeting agents with vast therapeutic advantages have been developed, but they often lead to multidrug resistance and adverse side effects. Thus, single-target therapy has drawbacks in the effective control of tubulin polymerization. Molecular hybridization, based on the amalgamation of two or more pharmacophores of bioactive conjugates to engender a single molecular structure with enhanced pharmacokinetics and biological activity, compared to their parent molecules, has recently become a promising approach in drug development. The practical application of combined active scaffolds targeting tubulin polymerization inhibitors has been corroborated in the past few years. Meanwhile, different designs and syntheses of novel anti-tubulin hybrids have been broadly studied, illustrated, and detailed in the literature. This review describes various molecular hybrids with their reported structural–activity relationships (SARs) where it is possible in an effort to generate efficacious tubulin polymerization inhibitors. The aim is to create a platform on which new active scaffolds can be modeled for improved tubulin polymerization inhibitory potency and hence, the development of new therapeutic agents against cancer.

## 1. Introduction

Cancer is one of the primary causes of mortality worldwide, consisting of more than 150 diseases categorized on the cellular level by increased cell proliferation and downregulation of pro-apoptotic pathways. Globally, the new cancer cases were diagnosed at ~19.3 million, with ~10 million cancer-caused mortalities in 2020. Meanwhile, an estimated 28.4 million new cancer cases (including nonmelanoma skin cancer (NMSC), except basal cell carcinoma) are predicted to occur in 2040 [1]. This is 47% higher than the 19.3 million resulting cases in 2020 if the estimated national rates in 2020 remain constant [2]. Figure 1 shows the global distribution of the top 13 cancer types with the highest mortality rate. Finding and developing new chemotherapeutic drugs to cure cancer ultimately is the goal of many medicinal chemists. However, the immediate goal is to find low-toxicity efficacious drug candidates, which is already an enormous challenge. There are several major classes of anticancer drugs, for example, (i) alkylating and intercalating agents (DNA-targeting drugs), (ii) topoisomerase inhibitors (iii) hormone therapeutics, (iv) antimetabolites (v) signal transduction inhibitors, and (v) mitotic spindle inhibitors. Chemical compounds that interfere with the assembly of microtubules play an essential role in cancer chemotherapy within these classes.

Microtubules are cylindrical protein polymers assembled in the cytoplasm of all eukaryotic cells by polymerization of *α**β*-tubulin heterodimers. Microtubules are characterized by a unique dynamic instability process, which involves continuously alternating phases of elongation and shortening interspersed with catastrophic disassembly events. In living cells, the minus ends of microtubules are associated with structures called microtubule organizing centers (MTOCs) [3]. The main MTOC in the cell is the centrosome, contiguously positioned to the nucleus. Microtubules play an essential function in the maintenance of cell shape and in the progression of cell division [4]. They are the principal components for the formation of the mitotic spindles. They all participate in signaling, intracellular transport, and cell motility [5]. Disturbance of microtubules may induce cell cycle arrest at the G2/M phase. Their pre-eminent importance in force generation in mitosis to enable the segregation of chromosomes makes microtubules attractive as intra-cellular targets for anti-cancer drug action.

Drugs that impede the mitotic spindle apparatus have been the mainstay of effective cancer chemotherapy. They are referred to as tubulin-binding agents, antimicrotubular, or anti-tubulin compounds (see Figure 2). They have drawn much interest in the past two decades, making microtubules important drug targets. Several natural compounds such as paclitaxel, epothilones, vinblastine, combretastatin, and colchicine act by altering microtubule dynamics. Besides, compounds originating from different medicinal chemistry strategies can be classified based on their mechanism of action and the location of the primary binding site. Tubulin polymerization inhibitors can interact with residues in either the colchicine binding site or Vinca alkaloids binding site, while the tubulin depolymerization inhibitors interact with the taxane site. These drugs have significantly improved survivorship rates and management of several cancer types among children and adults. Nevertheless, the intrinsic or acquired multidrug resistance (MDR) [6,7,8], as well as numerous adverse side effects such as chemotherapy-induced peripheral neuropathy, nausea, vomiting, diarrhea, constipation, paralytic ileus, urinary retention, bone marrow suppression, and neurotoxicity [9,10,11] of these drugs have been of significant concern. As a result, toxicity and drug resistance are the main properties requiring constant improvement when developing novel tubulin polymerization inhibitors with acceptable pharmacological profiles. Additionally, combination therapy has been a strategy aimed at enhancing the potency of several drugs such that one drug can impede the growth of cancer cells in a specific phase, and another cancer agent can function in a different phase. In addition to the complex regimens that use multiple medications, the combination of cancer chemotherapy with surgery reduces the number of cancer cells and radiotherapy destroys cancer cells even further. Hence, many cytotoxic hybrids have been clustered into regimens of anticancer behavior that improve treatment outcomes with fewer side effects. However, combination therapy is often non-advantageous due to poor patient adherence as well as the risk of overlapping toxicities.

Many medicinal chemistry researchers presently use the molecular hybridization approach in developing unique bioactive compounds with multiple molecular targets while being less susceptible to drug resistance. Triazole, tetrazole, benzimidazole, pyrazole, quinoline, quinazolinone, quinolinone, chalcone, coumarin, indole, and oxindole are attractive molecules with potent anticancer activities that have gained more attention; thus, they are used as building blocks for designing and developing novel chemical structures. This paper provides a comprehensive overview of the literature related to the hybridization of heterocyclic organic molecules, such as those mentioned above, targeting tubulin polymerization. We also attempt to shed light on structure–activity relations for the hybrid compounds and compounds with moderate activities, which are often overlooked.

## 2. Tubulin Polymerization Hybrids

### Triazole and Tetrazole Hybrids

Triazole occurs as an isomer depending on the position of the three nitrogen atoms in the ring. The two possible isomers include 1,2,3-triazole and 1,2,4-triazole. Triazole is highly versatile with a broad spectrum of biological activities. It has been introduced in several clinical drugs [12,13], including cancer drugs [14,15,16] (letrozole and anastrozole), thus drawing attention to the importance of this nucleus. Tetrazole is a useful building block commonly used in developing new drugs due to its metabolic stability, the non-metabolized bio-isosteric analog of carboxylic, cis-amide, and other expedient physicochemical properties. Furthermore, the isomer forms of tetrazole such as 1*H* and 2*H*-tetrazole have been reported to be effective for designing anti-cancer compounds. Combretastatin (CA-4) **1** has effectively inhibited tumor cell growth, such as MDR cancer cell lines, and displays significant inhibition of tubulin polymerization and excellent cytotoxicity against murine lymphocytic leukemia human ovarian and colon cancer cell lines. However, it lacks water solubility. On the other hand, 5F-203, **2** and its prodrug, phortress, are aqueous soluble and chemically stable, while the clinical evaluation of phortress displayed selective antitumor activity. Interestingly, Rao et al. [17]. synthesized cis restricted triazole/tetrazole analogue of CA-4 and benzothiazole scaffolds **3**. The *cis* configuration of the olefinic bond was restricted by incorporating triazole and tetrazole rings. The synthesized compounds were tested against selected human cancer cell lines, namely, prostate (DU-145), cervix (HeLa), lung adenocarcinoma (A549), liver (HepG2), and breast (MCF-7), using CA-4 as the reference compound. In the compounds with 1,2,4-triazole (**3**) derivatives, compounds **3a** and **3b** with a methyl substitution on the C′-3 position of the 2-phenylbenzothiazole moiety and OCH_3_ and F, substitution on the C′-6 position of benzothiazole moiety displayed great inhibitory activity with an *IC*_50_ value of 0.054 and 0.048 µM against the lung cancer cell line. These compounds also cause the arrest of cells in the G2/M phase and induce apoptosis in A549 cells by activation of caspase 3. They significantly inhibit tubulin assembly with *IC*_50_ values of 1.67 and 1.00 µM and bind to the colchicine site of beta (*β*) tubulin. The substitution of 1,2,4-triazole linker with 1,2,3,4- tetrazole (**4**) and 1,2,3-triazole (**5**) resulted in moderate activity. Figure 3 displays the most potent compound in the 1,2,3,4-tetrazole (**4a**) and 1,2,3-triazole (**3a**) group with *IC*_50_ values of 0.246 and 0.243 µM, respectively, against the lung cancer cell line. In summary, derivatives bearing phenyl ring attached to benzothiazole ring (R = Me) at C′3-position significantly increased the activity compared to other substituents. Based on the molecular docking, the pose view of **3b** fits well in the colchicine binding domain. The tri-OCH_3_ phenyl group has been found deep in the hydrophobic site of the *β* chain, and it is in close contact with amino acids, Val238, Thr239, Cys241, Leu242, Ala250, Leu252, Leu255, Ala316, Ala317, Val318, Thr353, Ala354, and Ile378, respectively. The benzothiazole is extended towards the *α*, *β* interface of the tubulin and forms close contact with Gln11, Ser178, and Tyr224 of the *α* chain and Glu247 and Gln248 residues of the *β* chain. However, 1-phenytriazole is in close contact with Asn101, Ala180, and Val181 residues of a chain and Asn258, Met259, and Lys352 residues of the *β* chain.

Xu et al. [18] synthesized series of 3,6-diaryl-7*H*-[1,2,4]triazolo[3,4-b][1,3,4]thiadiazines by replacing the (*Z*,*E*)-butadiene the spacer of vinylogous CA-4 **5** with a novel rigid [1,2,4]triazolo[3,4-b][1,3,4]thiadiazine moiety. Subsequently, the antiproliferative activity of the compounds was tested against three human cancer cell lines, including SGC-7901, A549 (gastric adenocarcinoma), and HT-1080 (fibrosarcoma) cells, using MTT assay and CA-4 was utilized as the positive control. Out of the 32 synthesized compounds tested for their antiproliferative activities. Compound **7a** showed the most powerful antiproliferative activities against SGC-7901, A549, and HT-1080 cell lines with an *IC*_50_ of 0.011–0.015 μM, compared to the standard drug CA-4 (0.009–0.013 μM). Besides, compound **7a** remarkably inhibited tubulin polymerization (*IC*_50_ = 1.6 µM), compared to CA-4 (*IC*_50_ = 0.92 μM). The substitution of 3-amino-4-methoxyphenyl (R_6_ = NH_2_, R_5_ = OCH_3_) and 3-hydroxy-4-methoxyphenyl (R_6_ = OH, R_5_ = OCH_3_) at the B-ring led to moderate activity. Meanwhile, the presence of 2,3,4-trimethoxyphenyl (R_1_ = R_2_ = R_3_ = OCH_3_) and 3,4,5-trimethoxy group (R_2_ = R_3_ = R_4_ = OCH_3_) at A-ring improved the activity. Tubulin polymerization affected by **8a** experienced inhibition and significantly disrupted microtubule dynamics. In addition, **8a** potently induced cell cycle arrest at the G2/M phase in SGC-7901 cells. Docking studies indicate that the compound binds to the colchicine binding site (*β* chain). A hydrogen bond formed between the oxygen atom of the OCH_3_ group of **8a** and the thiol group of Cys241. The nitrogen atom on the triazolo-thiadiazine linker forms a direct hydrogen bond with the residue of Ala250. The nitrogen atom of the amino group on the B-ring forms another hydrogen bond with the residue Asn349 (see Figure 4).

Piperazine has been reported as a potent compound that obstructs mitosis, induces cell apoptosis, binds to the colchicine site, and inhibits the tubulin [19,20]. Meanwhile, sulfonamide compounds are antibacterial, anticancer, antithyroid, antidiabetic, antiviral, and antihypertensive agents. In the published work of Manasa et al. [21], 1,2,3-triazole, sulfonamide, and piperazine moieties **9** were reported to be synthesized and evaluated for their in vitro cytotoxic potential against a selected panel of human cancer cell lines. These include BT-474 (breast), HeLa (cervical), MCF-7 (breast), NCI-H460 (non-small cell lung cancer), and HaCaT (human epidermal keratinocytes) using the MTT assay. The substitution of the R-group with *meta*-phenoxy and R_1_ with 4-Br showed improved cytotoxicity on the BT-474 cancer cell line (*IC*_50_ value of 0.99 ± 0.01 µΜ) among the synthesized compounds (see Figure 5). The tubulin targeting ability of **5** was moderate. It could also effectively inhibit tubulin polymerization with increased concentration and induce a blockade of the sub-G1 and G2/M phase in the cell cycle, including morphological changes and cell proliferation inhibition through the induction of apoptosis. The molecular modeling studies show the interaction of compound **9a** with tubulin via the colchicine binding site. The sulfonyl group has been shown to establish a strong hydrogen bond interaction with Ser178, and the triazole moiety forms two hydrogen bond interactions with binding site residues (Ala250 and Lys254). It shares hydrophobic contacts with residues of *α*/*β*-tubulin and fits well into the binding site interphase of *α*/*β*-tubulin (see Figure 5).

Wu and colleagues [22] designed new molecular hybrids (Figure 6) composed of sub-units similar to XRP44X **10**. The authors replaced the pyrazole ring of XRP44X with another heterocycle ring. XRP44X can effectively obstruct FGF2-induced activation of the Ras-Net (Elk-3) signaling pathway, inhibiting tumor growth and metastasis, and reducing toxicity [23,24]. The inhibition of tubulin polymerization and the morphology of the actin cytoskeleton by XRP44X also contribute to its importance. The antiproliferative activity of the synthesized compounds was tested in vitro using three human cancer cell lines. Namely, SGC-7901, A549 cells, and HT-1080 (fibrosarcoma) cells were used. Compound **12a** (R_1_ = 2-chloro) exhibited significant antiproliferative activity with an *IC*_50_ value between 5 and 52 nM towards all three cell lines, compared to CA-4 (7–48 nM). The structure–activity relationship determined that the positioning of the substituent and electronegativity has minimal effects on the activity. The compound exerts its anti-cancer activity by inhibiting tubulin polymerization (*IC*_50_ = 2.06 μM) and binding at the colchicine binding site (see Figure 6).

Based on the properties of XRP44X as anti-angiogenic and anti-tubulin, Wang et al. [25]. used the same approach similar to Wu et al. [22] by replacing the pyrazole group XRP44X with moiety bearing hydrogen bonding acceptor. The compounds were evaluated screened against SGC-7901 (gastric), A549 (lung), and HeLa (cervical) using MTT assay. Among the targeted compounds, compound **13a** (R_1_ = H, R_2_ = 3,5-dimethoxy) displayed the most potent antiproliferative activity against HeLa cell lines with an *IC*_50_ value of 0.403 ± 0.02 μM. Compound **13a** suppressed microtubule assembly and disrupted the cytoskeleton in a fashion similar to that of CA-4. Compounds with a halogen group substituent on the A-ring demonstrated slight antiproliferative activities. Substituents on the A-ring displayed an arrangement of influence according to: *ortho*- > *meta*- > *para*-substituted, respectively. There was a decrease in antiproliferative activities when substituents were placed in position 4 of the A-ring. The position of the substituents on the D ring is the major factor affecting the activity. The molecular docking studies showed that the 1*N* of 1,2,4-triazole of **13a** interacts via a crucial hydrogen bond with the amino acid residues Lys*β*352, which was not quite common for XRP44X, hence suggesting that it may exert its biological activities by binding to the colchicine site (see Figure 6).

Sulfanilamide skeleton is a bioactive unit that exhibits strong anti-tumor activity [26,27]. Due to the anti-tumor activity of sulfanilamide and triazole, Guo et al. [28] projected that a sulfanilamide **14** ring linked to a 1,2,3-triazole **15** unit would serve as novel and effective tubulin polymerization inhibitors. The synthesis and antiproliferative activity of sulfanilamide-1,2,3-triazole hybrids **16** against three selected human cancer cell lines, namely, BGC-823 (gastric), MGC-803 (gastric carcinoma), and SGC-7901, were reported. Compound **16a** showed the most excellent inhibitory effect against MGC-803 cells, with an *IC*_50_ value of 0.4 ± 0.1 μM. The substitution of 7-hydroxy-2*H*-chromen-2-one with *N*-heterocycle functional groups such as indoline, 6-methoxyindoline, 1,2,3,4-tetrahydroquinoline, and 2,3,4,5-tetrahydro-1*H*benzo[b]azepine led to a decrease in the antiproliferative activity with *IC*_50_ of 10.5 ± 1.3, 8.7 ± 0.4, 3.8 ± 0.3, 12.8 ± 0.2 μM, respectively. The in vitro tubulin polymerization assay showed that **16a** is a tubulin polymerization inhibitor with an *IC*_50_ value of 2.4 μM (see Figure 7).

Podophyllotoxin (PPT) **17** is a non-alkaloid toxin lignin isolated from *Podophyllum peltatum* L. and *Podophyllum hexandrum*, and it possesses a large variety of medical applications. Etoposide and teniposide are semi-synthetic derivatives of podophyllotoxin and differ significantly in their mode of action. These compounds are DNA topoisomerase II inhibitors, and the inhibition of topoisomerase II blocks DNA from separating. Podophyllotoxin can successfully impede the microtubules construction. Their shortcomings include a lack of water solubility and drug resistance. The modifications of C-4 position have shown excellent inhibitory activity. In this regard, a series of triazolo-4*β*-amidopodophyllotoxin derivatives using an amide spacer were synthesized and examined for their cytotoxic activity against four human cancer cell lines, namely MCF-7 (breast cancer), B16 (oral cancer), HT 29 (colon cancer), and HeLa (cervical cancer) as reported by Vishnuvardhan et al. [29]. Podophyllotoxin and etoposide were employed as the reference compounds. Compound **18a** (*IC*_50_ = 0.9 µM) and **18b** (*IC*_50_ = 0.07 µM) exhibited promising anticancer activity compared to podophyllotoxin (*IC*_50_ = 2.36 µM) and etoposide (*IC*_50_ = 2.34 µM) in the HeLa cell line. Furthermore, when substituents such as toluene, bromobenzene, methoxynaphthalene, chlorobenzene, and pyridine were inserted in the R position, decreased cytotoxic activity was observed. This is observed for all the cancer cell lines compared to podophyllotoxin and etoposide. The conjugates **18a** and **18b** inhibited tubulin polymerization by 46.40% and 48.50%, respectively, compared to control podophyllotoxin that showed 50.19% inhibition. They induced apoptotic cell death by targeting tubulin leading to cell cycle arrest at the G2/M phase followed by upregulation of the caspase-3 activity. The docking studies revealed that both compounds bind at the interface of the *α*/*β* chains of the tubulin dimer (see Figure 8).

A combination of the structure of benzo [b]furan, **19**, **20**, and triazole **21** to produce a single molecule has been explored. The hybrid compounds containing benzo[*b*]furan and triazole moieties (**22**) with their antiproliferative activities against four-panel of human cancer cell lines (HCT116 colon cancer, HepG2 hepatic carcinoma, HeLa human epithelial cervical cancer, and A549 non-small-cell lung cancer) were reported by Qi et al. [30]. The 6-methoxy bearing ring’s antiproliferative activities were superior to the unsubstituted compounds. Compound **22a** exhibited the most excellent antiproliferative activities against all the tested cell lines, with *IC*_50_ values of 0.87 ± 0.79 (HCT116), 0.73 ± 0.67 (HeLa), 5.74 ± 1.21 (HepG2), and 0.57 ± 0.31 µM, respectively. This indicated the importance of the 6-methoxy group for excellent antiproliferative activity. Meanwhile, substituting ethoxy and isopropoxy groups at the R_2_ position led to a decrease in antiproliferative activities. The displacement of aniline by cyclopropylamine, piperidine, and morpholine at position R_2_ gave inferior cytotoxic activities. Compound **22a** inhibited tubulin polymerization in a dose-dependent manner with *IC*_50_ value 4.1 ± 0.1 μM, comparable to CA-4 (1.0 ± 0.1 μM). The promising compound (**22a**) disrupted microtubule organization blocked the cell cycle in the G2/M phase by affecting cycle-related proteins. This led to apoptosis by downregulating anti-apoptotic proteins and reducing the MMP of A549 cells. Molecular modeling suggests that 3,4,5-trimethoxybenzene ring of **22a** fits well into the *β*-tubulin and forms a hydrophobic interaction with amino acid Cys241, while the benzo[*b*]furan ring has been seen fronting *α* tubulin and the 2-*N*-phenyl amide is positioned close to Ser178 and Val181 (see Figure 9).

A novel series of tubulin inhibitors bearing an indole-1,2,4-triazole (**25**) scaffold have been synthesized, and their biological activity has been reported [31]. Among the 18 compounds evaluated for anti-proliferation activity against HepG2 (human hepatoma cells), HeLa (human cervix cell), MCF-7 (breast cancer cell), and A549 (human lung cancer cell), compound **25a** displayed excellent activity with *IC*_50_ values of 0.23 ± 0.08, 0.15 ± 0.18, 0.38 ± 0.12, and 0.30 ± 0.13 μM, respectively. The substitution of electron-donating groups such as OCH_3_ and CH_3_ (**25a**–**d**) enhanced the anti-proliferative activity compared to the electron-withdrawing groups (F, Cl, Br, I, and CF_3_). The results from the tubulin polymerization assay showed that compound **25a** was the most potent anti-tubulin agent (*IC*_50_ = 2.1 ± 0.12 μM), superior to the inhibitory activity of colchicine (2.52 ± 0.23 μM). Moreover, **25b**–**d** inhibited tubulin polymerization with *IC*_50_ values of 5.08 ± 0.16, 3.4 ± 0.43, and 6.92 ± 0.21 μM respectively. **25a** induced cell cycle arrest at the G2/M phase associated with a reduction in mitotic cells and induced downregulation of the anti-apoptotic proteins Bcl-2, Bcl-x1, and mainly Mcl-1, an essential component for cell survival. Molecular docking revealed that compound **25a** interacts with tubulin at the colchicine binding site (see Figure 10).

Prasad and colleagues focused on replacing the sulfonamide group in the E-7010 **26** scaffold with triazole-linked carboxamide [32]. E-7010 was found to arrest the cell cycle and caused apoptosis in the G2/M phase and show cytotoxic activity by inhibiting tubulin polymerization [33]. Besides, the compound exhibited remarkable in vivo antitumor activity against rodents infected with tumors and human tumor xenograft [34]. In addition, the presence of a pyridine ring with 2-anilino and 3-sulfonamido enhances aqueous solubility and toxicity, and this is the only chemical agent that binds at the three major binding sites on *α*-,*β*-tubulin subunits. The synthesized compounds **27** by Prasad et al. were screened against four human cancer cell lines for their anti-proliferative activity. Their effort produced **27a** (*IC*_50_ values of 1.59, 1, and 1.34 μM, R_1_ = 3,5–diOCH_3_, and R_2_ = 3-OPh) which is more potent than E7010 (*IC*_50_ = 1.71, 1.62, and 2.188 in DU-145, A-549, and HeLa cell lines. In addition, good tubulin polymerization inhibition (*IC*_50_ value 2.04 μM) was found compared to E7010 (*IC*_50_ value 2.15 μM). The cytotoxicity effect in the presence of electro-donor and acceptor groups on ring A with the same substituents on ring D was observed as; 3,5–diOCH_3_ > 3,4–diOCH_3_ > 4–OCH_3_∼2,4–diOCH_3_ > 3,4,5–triOCH_3_ > 4–H > 4–OH > 4–F, whereas changing of substituents on ring D with constant substituent on ring A, led to reduce cytotoxicity in the following order 3–OC_6_H_5_ > 3– OCH_3_ > 4– OCH_3_ > 3,4,5–tri OCH_3_ > 4–H. In addition, the introduction of electron donors on both rings is crucial for the increase in cytotoxicity activity. The authors reported that **16a** fit well into the colchicine binding site in the *β*-subunit at the *α*/*β* interface of the tubulin. In addition, the active compound displayed the same binding landscape as E-7010. The 1-(3-phenoxybenzyl)-1*H*-1,2,3-triazole-4-carboxylic acid and methoxybenzene sulfonamide moieties extend towards the *α*-subunit. Moreover, the *N*2-substituted phenylpyridine-2,3-diamine and pyridine-2,3-diamine motifs fit into the *β*-subunit. Compound **16a** formed hydrophobic interactions with the side chains and peptide backbones of the *β*-subunit (see Figure 11).

The pharmacophoric imidazopyridine **29** moiety have been found to have a wide range of biological activities. This scaffold has been explored to design new hybrids of antimicrobial, anticonvulsant, antipyretics, anti-inflammatory, anticancer, and many others. Imidazopyridine-guanylhydrazone displayed strong anticancer activity against different cancer cell lines. The derivative also arrested cell cycle in G2/M phase on SK-LU-1 cell line, downregulate cyclin-D1, E1, CDK2, and caused activation of Caspase-3 [35,36]. Because of this, Sayeed and co-workers combined triazole **30** and imidazopyridine **29** group into a single entity [37]. They were subsequently evaluated for their cytotoxic effect in human cancer cells, precisely, lung (A549), prostate (DU-145), colon (HCT-116), and breast (MDA-MB 231). Two conjugates, **31a** and **31b** (*IC*_50_ = 0.51 and 0.63 μM) were found to be more active than the remaining conjugates against A549 [37]. These conjugates caused cell cycle arrest at the G2/M phase, signifying inhibition of tubulin polymerization and induced apoptosis through the mitochondrial pathway. The order of potency of the substitution on ring B is H > Cl > OCH_3_. Whereas the ring C with triOCH_3_ substituent displayed the highest potency among the electro-donating group, the Cl atom showed superior potency than other electro-withdrawing groups (see Figure 12).

Wang et al. [38] proposed a series of novel XRP44X analogs **33** by replacing the pyrazole ring of XRP44X **10** with 1*H*-tetrazol, (hydrogen-bonding acceptors). All the compounds were evaluated for anti-cancer activity followed by anti-tubulin activity. Among the 32 compounds synthesized and evaluated, **33a** exhibited excellent activity against three cancer cell lines, mainly SGC-7901(gastric), A549 (lung), and HeLa (cervical), with *IC*_50_ values of 0.090, 0.650, and 0.268 μM, respectively. The presence of substituent into the ortho-position of A-ring significantly enhanced the antiproliferative activity, and the order of substituent is as follows: 2-methyl > 2-fluoro > 2-chloro > H. While the presence of 3,5-diOCH3 at the D-ring remarkably improved the anticancer activities. Notably, pharmacokinetic properties of 18a were projected to be better than XRP44X. Further, **33a** inhibited tubulin polymerization and disrupted microtubule in SGC-7901 cells. The cell cycle analysis indicated that **33a** prevented SGC-7901 cell in the G2/M phase and induced apoptosis in a dose and time-dependent. In the molecular dockings study, **33a** interacted with Ala*β*31. H-bonding was observed between the residue of Asn*β*258, Lys*β*352, 2*N* of 1*H*tetrazole, and the 4*N* of 1*H*-tetrazole of compound **33a.** In addition, H-bonding forms between the carbonyl group of XRP44X or **33a** with amino acid residue Alab317 (see Figure 13).

## 3. Benzimidazole Hybrids

Benzimidazole is a privileged heterocyclic scaffold found in many natural and pharmacologically functional molecules. Benzimidazole, also known as 3-azaindole, azindole, benziminazole, benzoglyoxaline, 3-benzodiazole, and 1,3-diazaindene; containing fusion of benzene and imidazole. The organic molecule is colorless, freely soluble in alcohol, melts at 170.5 °C, and boils at 360 °C. The core structure of benzimidazole is a vital pharmacophore at present. It has been used as a preferred scaffold for synthesizing selected drugs of interest in medicinal chemistry, including antimicrobial [39,40], analgesic [41], antihelmintic [42,43], antioxidant [44], antimalarial [45], antiviral [46], and anticancer [47] activity. Many benzimidazole-based derivatives are available as medicines such as bendamustine, veliparib, nocodazole, liarozole, and pracinosta. Further, dehydroabietic acid is a natural diterpenic resin acid, and its derivatives exhibited strong anticancer activity through different mechanisms of action **38**. The 3,4,5-trimethoxyphenyl (TMP) **1**, **39**, **40**, and benzimidazole **34**–**37** sub-unit were a remarkable pharmacophoric group of tubulin inhibitors. The units can be combined to generate a new series of dual-acting action hybrids. Considering the biological importance of benzimidazole, dehydroabietic, and the compounds containing the trimethoxyphenyl group, a series of dehydroabietic–benzimidazole conjugates (**41, 42**) were synthesized by Miao et al. [48]. The synthesized compounds were further evaluated for their in vitro cytotoxic activity in human cancer cells, namely, SMMC-7721 (hepatocarcinoma), MDA-MB-231 (breast), HeLa (cervical), CT-26 (mouse colon cancer), and QSG-7701 (normal human hepatocyte). Most of the compounds showed significant anti-cancer activity. Among the tested compounds methyl 2′-(1*H*-indol-5-yl)-12-bromo-13,14-imidazolyldeisopropyl dehydroabietate (**41a**) possessed *IC*_50_ values of 0.08 ± 0.01, 0.19 ± 0.04, 0.23 ± 0.05, and 0.42 ± 0.07 µM, against SMMC-7721, MDA-MB-231, HeLa, and CT-26 cancer cell lines, and significantly decreased cytotoxicity against the normal hepatocyte cell line, QSG-7701 (*IC*_50_ value = 5.82 ± 0.38 µM). The substitution of 12-Br with 12-(3,4,5-trimethoxyphenyl) moiety to produce **42** led to a decrease in the anti-cancer activity of the compounds. This observation indicates that substituting 12-Br with 12-(3,4,5-trimethoxyphenyl) moiety will not improve the anticancer activity. In addition, the nature of the aryl groups on benzimidazole rings influenced the cytotoxic activity of the tested compounds. When the Br atom is at the twelfth position, the fluoro or Cl groups on the *para* position enhanced the cytotoxicity compared to fluoro or Cl atom on the *meta* or *ortho* positions. Compared to their counterpart, derivatives with 1*H*-indol-5-yl and 1*H*-indol-3-yl functional groups showed excellent inhibitory activities. Compound **41a** displayed stronger potency than colchicine at a 10 µM concentration in the tubulin polymerization assay. In summary, **41a** arrest SMMC-7721 cells at the G2/M phase in a dose-dependent manner, inhibit the tubulin polymerization, and disorganize the cancer cells’ microtubule network (see Figure 14).

ABT-751 [33] **43** and HMN-214 [49] **44** are sulfonamide-bearing compounds with strong anti-tubulin and antimitotic properties. Due to the interesting properties of sulfonamide **42**–**44** and benzimidazole, Wang et al. [50]. focused on synthesizing a new series of 1-phenylsulphonyl-2-(1-methylindol-3-yl)-benzimidazole derivatives (**45**). The compounds were further probed for anti-cancer against human cancer cell lines, specifically, A549 (lung), Hela (cervical), HepG2 (liver), and MCF-7 (breast). In addition, the anti-tubulin activity was further examined. The cell proliferation assay results showed that most synthesized compounds exhibited notable activity for cancer cells, specifically A549. Compounds **45b**, **45c**, and **45e** showed notable antiproliferation activity than the positive control CA-4 for A549, HeLa, HepG2, and MCF-7, respectively, as shown in Figure 15. It was found that compounds **45a**–**e** have ameliorated antiproliferation activity compared to the positive control colchicine. The SAR studies suggested the presence of electron-donating substituents (OCH_3_ and CH_3_) at either R_1_, R_2_, and R_3_, and the bulkiness of the electron-donating substituents such as two, three, or more OCH_3_ at position R_3_ could enhance the tubulin inhibitory activity. The order of antiproliferative activity substituent is introduced to R_2_ position is as follows: CH3 > H > Br. The in vitro tubulin polymerization assay showed that **45b** exhibits equivalent inhibitory activities to CA-4 (*IC*_50_ = 1.9 μM), while **45c** and **45e** displayed improved inhibitory activities on tubulin polymerization compared to those of CA-4. Moreover, **45a**–**e** exhibited better inhibitory activity than colchicine (*IC*_50_ = 2.54 μM). Notably, **45c** exhibited the most antiproliferative and anti-tubulin activity (see Figure 15, Table 1).

This mechanism of action for new antiproliferative and anti-tubulin agents was also explored in the design of a new series of benzimidazole, based on the amalgamation of different pharmacophores as shown in Figure 16. Twenty-five novel hybrids of benzimidazole combined with pyrazole **48** were reported by Wang et al. [51]. The compounds were evaluated against four cancer cell lines. Among the tested compounds from the benzenesulfonamide hybrids, **48a**–**c** showed excellent antiproliferative activities compared to colchicine **39**. Moreover, **48c** was observed to have superior activity than CA-4 **1**, as shown in Table 2. The SAR showed that CF_3_ substituent at the *para* position of the A-ring and OCH_3_ and CH_3_ substituent on the B-ring enhances the anti-cancer activity. Compound **48c** was the most active compound against cancer cell lines with *IC*_50_ values ranging from 0.15 to 0.33 μM. In contrast, compounds **48a**–**d** exhibited better anti-tubulin activities than colchicine (*IC*_50_ = 2.26 μM). Compound **48c** was observed as the most effective compound for tubulin polymerization with an *IC*_50_ value of 1.52 μM (see Figure 16, Table 2). These findings showed that benzenesulfonamide-bearing pyrazole ring derivatives could be a new structural template for developing promising antitumor agents acting on tubulin polymerization.

The anti-mutagenicity effect of cinnamic acid lies in its ability to network with cellular nucleophiles such as glutathione (GSH) and cysteine. At the same time, the existence of a,ß-unsaturated carbonylic functional unit is liable for cellular interactions of cinnamic acid [52]. Donthiboina et al. [52] reported a synthetic route to benzimidazole and cinnamic acid hybrids **51** using a molecular hybridization approach, and the synthesized compounds were screened for their cytotoxic activity. Among the tested compounds, **51a** displayed potent cytotoxic activity against lung cancer cell line with *IC*_50_ value of 0.29 ± 0.02 μM and showed less cytotoxicity when tested on NRK-52E (normal rat kidney epithelial cell line) with *IC*_50_ value of 1.58 ± 0.43 μM. The SAR showed a lack of substitution on the phenyl ring of cinnamides (R_1_ = H), resulting in a reduction in cytotoxicity. The insertion of electron-withdrawing groups (R or R_1_ = F, Cl, Br, and NO_2_), on the phenyl ring of benzimidazole or cinnamide led to either reduced or lost activity. In contrast, electron-rich substitution on either of the rings resulted in moderate to good activity. Besides, substituting 3,4,5-tri OCH_3_ on the phenyl ring of cinnamide (**51b** and **51c**, *IC*_50_ values >30 μM) reduced the activity due to steric hindrance. Compound **51a** induced apoptosis by producing ROS, inhibited tubulin polymerization (*IC*_50_ value = 4.64 ± 0.09 μM), and changed the cell cycle by blocking progression at the G2/M phase (see Figure 17).

A series of thioether and amine-bridged cinnamide derived pyrimidine-benzimidazole hybrids were designed based on the pharmacophore hybridization approach by mimicking the structural similarities of reported anti-tubulin agents [53]. The SAR analysis showed that substitution featuring bulkier groups such as 2-bromo-4,6-dimethoxy and 2-bromo-4- methoxy on the cinnamide scaffold were stable in both the molecular series (**58a**, **58b**, **58c**, **59a**, and **59b**). Among the 31 compounds tested for anti-cancer activity, **59b** with CF_3_ group at position two of benzimidazole has the highest potency, as shown in Figure 18 and Table 3. Additionally, the hybrids with amine spacer (**59**) afford improved antiproliferative activity in A549 cancer cell lines. The presence of electron-donating (OCH_3_), electron-withdrawing (OCHF_2_), or the absence of substitution in the thioether moieties (**58**) slightly induces the antiproliferative activity. The anti-tubulin analysis using immunofluorescence staining showed that **59b** (*IC*_50_ value = 5.72 ± 0.51 μM) inhibited microtubule organizations in a concentration-dependent mode, immobilized the cells at G2/M phases of the cell cycle, and enhanced ROS assembly. Finally, molecular docking studies of the active compound revealed that cinnamide fits well into the *β*-hydrophobic pocket comprising residues Tyr202, Val238, Cys241, Leu242, Leu255, Ala316, Ala317, Val318, Ala354, and Ile378, respectively. Furthermore, the benzimidazole ring was found to deeply penetrate the *α*-tubulin T5-loop (see Figure 18, Table 3).

The presence of a cyano group in many reported benzimidazole bearing 2,3-disubstituted acrylonitriles as well as their cyclic analogues strongly contributed to the high antiproliferative activity observed from the compounds [54,55]. Based on this panorama, Perin and colleagues [56] designed a new series of *N*-substituted-2-benzimidazolyl acrylonitriles hybrids using a molecular hybridization approach. They evaluated their antiproliferative activity towards eight human cancer and one non-cancerous cell line. Among the tested compounds, **62a** and **62b** showed superior activity in all the cancer cell lines compared to the standard compound. Notably, introducing a cyano group at the benzimidazole scaffold resulted in lower inhibitory activity. The *N,N*-dimethylaminophenyl ring and the isobutyl side chain inserted into the *N*-position of benzimidazole moieties enhanced the antiproliferative activity. The author further examined the mechanism of the action of the most active antiproliferative agents, their toxicity, and their anti-tubulin properties. Notably, the active compounds are less toxic than docetaxel and staurosporine and disrupt the microtubule network, which is comparable to the mode of action of vincristine (see Figure 19, Table 4).

## 4. Pyrazole Hybrids

Pyrazoles are diazoles with a five-membered ring, useful in organic synthesis, the most studied compounds among the family of azoles. In particular, the heterocyclic cycle has a diverse range of chemical and biological properties [57,58,59] that can be traced back to several well-established therapeutic agents in different classes. Because of the interesting pharmacological properties of pyrazole hybrids, they garnered much attention. The newly synthesized derivatives of pyrazole-naphthalene (**65**) and their in vitro cytotoxicity have been reported [60]. The placement of an electron-donating group at position four of the phenyl ring favors antiproliferative activity. The results of introducing electron-withdrawing groups (R = 4-Cl, 3-Cl, 2-Cl, 3-F, 4-F, 2-F, 3-Br, and 2-Br) into the phenyl ring led to an insignificant increase in the inhibitory activity. Compound **65a** with *IC*_50_ = 2.78 ± 0.24 μM bearing ethoxy group (R=4-C_2_H_5_O) at the position four of the phenyl ring, was found to be five-fold superior to the standard drug cisplatin (*IC*_50_ = 15.24 ± 1.27 μM). Compound **65a** was further identified as a new potent inhibitor of tubulin assembly by inhibiting tubulin polymerization with the *IC*_50_ value of 4.6 μM (see Figure 20).

(3,4,5-trimethoxyphenyl) pyrazolo[3,4-*b*]pyridines (**68**, **69**) were generated from two-step Suzuki coupling reactions and tested for antiproliferative activities by Jian et al. [61]. The 3,4,5-trimethoxyphenyl at position-3 favored the antiproliferative activity. The compound with 3-hydroxy,4-methoxy phenyl group (**69a**, R = 3-OH,4-CH_3_OC_6_H_3_) was found to exhibit superior inhibitory activity against all the four tumor cell lines; specifically, MCF-7 (human mammary adenocarcinoma cells), MDA-MB-231 (human breast cancer cells), HeLa (human cervical cancer cells), and Kyse150 (human esophageal squamous cells) with *IC*_50_ values of 27.22 ± 2.31, 27.04 ± 6.42, 18.08 ± 1.48, and 62.82 ± 2.52 μM, respectively. The electron-withdrawing groups at the 4-position (R=4-FC_6_H_4_, 4-ClC_6_H_4_, 4-CNC_6_H_4_, and 4-CH_3_CO_2_C_6_H_4_) of the phenyl moiety resulted in inferior antiproliferative activity. **69a** effectively arrested HeLa cells in the G2/M phase. In addition, compound **69a** exhibited excellent anti-tubulin activity with a 31% (10 µM) inhibition of tubulin polymerization. The molecular docking showed that **69a** fits well into the colchicine binding site at the interface of *α*/*β*-tubulin (see Figure 20).

## 5. Quinoline Hybrids

Pharmacological studies have shown that quinoline’s annulus system has many biological activities. The marketed clinically-approved drugs containing the quinoline core include bedaquiline for multidrug-resistant tuberculosis therapy [62] and irinotecan used to treat colorectal cancer [63]. Quinoline-based compounds play a vital role in the development of anti-cancer drugs. They have been very successful with different mechanisms of action, such as growth inhibition by cell cycle arrest, apoptosis, angiogenesis inhibition, cell migration disruption, and modulation [64]. The quinoline hybrids with a tubulin polymerization-disrupting mechanism are described in this review. In the published work of Abdelbaset et al. [65], in order to improve the aqueous solubility challenge of CA-4 **1** through hydrogen bonding, the authors introduced a quinoline hybrid to replace the trimethoxyphenyl rings. Subsequently, quinoline–furanone (**70**), quinoline–pyrrolone (**71**), and quinoline–pyridazinone (**72**) conjugates were synthesized. The compounds were screened in one-dose evaluation for their cytotoxic activity against a panel of 60 cell lines adhering to the NCI approach. Quinoline–furanone hybrids **70** exhibited low anti-cancer activity against a different panel of cancer cell lines, while pyrrolones and pyridazinones exhibited promising anti-cancer activity. Compound **71b** (R = H, R_1._ =SCH_3_) showed anti-cancer activity against all cancer cell lines with *IC*_50_ values of 2.0, 2.6, 1.9, and 1.9 μM against pancreatic (panc-1, and paca-2,), lung (H-460), and colon (HT-29), respectively. The insertion of the 2-hydroxyl group (R_1_ = OH) into the quinoline structure reduces the inhibitory activity, while substituting 2-methoxy (R_1_ = OCH_3_), or 2-methylthio (R_1_ = SCH_3_) moiety enhances the activity. In addition, **71a** (R = H, R_1_ = OCH_3_), **71b**, and **72a** (R = CH_3_, R_1_ = OCH_3_) showed inferior EGFR inhibitory activity with *IC*_50_ values of 12.5, 10.6, and 12.5 μM and superior BRAF inhibitory activity with *IC*_50_ values of 3.4, 1.9, and 2.9 μM, respectively. The pyrrolone series, specifically **71c** (R = CH_3_, R_1_ = SCH_3_), exhibited remarkable antiproliferative activity with moderate selectivity against CNS and renal cancer, selectivity ratio = 3.49 and 3.56, respectively. Compounds **71b** and **72a** displayed superior tubulin polymerization inhibitory effect of 988 and nM 1002 nM, respectively, compared to vincristine (805). It can be concluded that quinolinyl pyrrolone compounds can serve as a template for developing antiproliferative agents with tubulin polymerization inhibitory activity. Which can also display pre-G1 apoptosis and cell cycle arrest at the G2/M phase. The visual detail of the molecular docking study showed that **71b** binds deep into the hydrophobic pocket and forms interactions with the following residues: Ala250, Leu255, Lys254, Ala316, Ala254, Val318, Leu248, and Ala180, respectively (see Figure 21).

A series of novel quinoline–indole derivatives were synthesized by replacing 3,4,5- trimethoxyphenyl and isovanillin moieties of isoCA-4-bearing quinoline and indole rings by Li and et al. [66]. The synthesized compounds were evaluated for their anticancer and anti-tubulin activity. The cancer cells used in their study include human hepatocellular carcinoma (HepG2), epidermoid carcinoma of the nasopharynx (KB), human colon cancer cells (HCT-8), human breast cancer cells (MDA-MB-231), mouse liver cancer cells (H22), and human normal hepatocytes (LO2). Compounds bearing methyl **78a** and hydroxymethyl **78b** were more potent than the other groups against K562 cells and the five cancer cell lines with *IC*_50_ values ranging from 5 to 11 µM, which are comparable to the activity observed in CA-4. The active compounds (**78a**, R_1_ = CH_3_, **78b**, R = CH_2_OH) exhibited tubulin polymerization at *IC*_50_ values of 2.54 and 2.09 µM, which is comparable to that for CA-4 (*IC*_50_ = 2.12 µM). It was found that compound **78a** disintegrates microtubule assembly and arrests the cell cycle at the G2/M phase, inducing apoptosis and depolarizing mitochondria of K562 cells in a dose-dependent manner. In addition, the in vivo antitumor activity of **78a** and **78b** were corroborated in the H22 liver cancer xenograft mouse model. The result showed that **78a** was able to suppress tumor volume and abridged tumor weight by 63.7% at the 20 mg/kg daily dose (i.v.). without obvious toxicity, which was more effective than CA-4 and CA-4P. (see Figure 22).

Mirzaei et al. combined the pharmacophores of 2-aryltrimethoxyquinoline **79** and chalcone **80** to design and synthesize new promising anti-cancer agents and tubulin polymerization inhibitors [67]. Four human cancer cell lines including A-2780 (human ovarian carcinoma), A-2780/RCIS (cisplatin-resistant human ovarian carcinoma), MCF-7 (human breast cancer cells), and MCF-7/MX (mitoxantrone resistant human breast cancer cells), and normal Huvec cell line (human umbilical vein en-dothelial cells) were employed in the MTT assay. Among the forty-eight compounds synthesized and evaluated against the four panels of cancer cell lines, compound **81a** bearing dimethoxyphenyl group (R_1_ = R_2_ = OCH_3_) on ring C and 4-nitrophenyl (R_6_=NO_2_) on ring D showed excellent cytotoxic effects against A-2780, A-2780/RCIS, MCF-7, and MCF-7/MX cancer cells (see Table 5). At the same time, **81b** (R_1_ = R_2_ = R_3_ = OCH_3_, R_6_ = NO_2_) and **81c** (R_1_ = R_2_ = R_3_ = OCH_3,_ R_6_ = R_7_ = OCH_3_) showed superior cytotoxic activity on A-2780 cancer cells compared to other derivatives. Meanwhile, trimethoxyphenyl (ring C) substitution with dimethoxy phenyl, 2-methoxyphenol, or 4-methoxy phenyl diminishes the cytotoxic activity. In addition, the substitution of the methoxy group of the ring C with bulky groups, namely phenoxy (R_6_ = OPh) or benzoyl at either the R_4_ or R_7_ position, reduced the cytotoxic effect. Notably, substitutions on ring D enhanced the cytotoxicity activity; this includes insertion of the nitro group at the para position of ring D. Compounds **81a** and **81c** inhibited tubulin polymerization by 45.62% and 40% at a concentration of 20 μM, respectively. **81c** arrested the cells in the G2/M phase, reduced cells accumulation in the G2/M phase, and enhanced the number of apoptotic cells (cells in the sub-G1 phase) (see Figure 23, Table 5).

A novel series of quinoline conjugates of combretastatin A-4 incorporating rigid hydrazone and cyclic oxadiazole spacers were synthesized [68]. The synthetic strategy involved, (i) bioisosterically replacement of CA-4 **1** (ring B) with quinolyl and 3,4,5-trimethoxyphenyl, and (ii) introduction of the hydrazone open linker and its cyclic form, the oxadiazole ring in place of the cis-olefinic bond for enhancement of structural rigidity. The compounds were evaluated for their antiproliferative and tubulin polymerization inhibitory properties. Four representative cancer cell lines, specifically MCF-7 (breast), HL-60 (leukemia), HCT-116 (colon), and HeLa (cervical) cancer cells, were used in the MTT assay. In the quinoline—hydrazone hybrids, **82a** (7-*O*-isopropyl) emerged as the most potent compound with *IC*_50_ values of 0.040, 0.026, 0.022, and 0.038 µM, respectively, in HL-60, MCF-7, HCT-116, and HeLa cancer cells. The presence of methyl substituent on different positions of the quinoline ring affected the antiproliferative activity. Compound **82b** with a methyl group at position six of the quinoline ring displayed notable potency comparable with CA-4 against all four cell lines (*IC*_50_ values range between 0.028–0.371 µM) and compared to 7-methyl and 8-methyl analogs. Meanwhile, in the series of the quinoline–oxadiazole hybrids, **83a** (R = 7-methyl) displayed superior antiproliferative activity with *IC*_50_ values ranging from 0.012–0.210 µM. Compound **83a** exhibited the most potent tubulin polymerization inhibitory activity with *IC*_50_ of 1.32 µM, which corroborates the antiproliferative inhibitory activity (*IC*_50_: 0.022–0.040 µM) followed by **82b** (R = 6-methyl) with *IC*_50_ values of 1.48 µM. Moreover, among the quinoline–oxadiazole derivatives, the **83a** (R=7-methyl), **83b** (R=7-*O*-isopropyl), and **83c** (R=7-benzyloxy) substituents showed effective inhibition of tubulin assembly with *IC*_50_ values of 2.41, 2.31, and 2.29 µM, which is comparable to CA-4 (*IC*_50_ value = 2.17 µM). The potent quinoline compounds with the lowest inhibitory activity showed approximately the same ability to inhibit tubulin polymerization as CA-4. The substitution of hydrazone linker enhanced the synthesized compounds’ inhibitory activity and their tubulin polymerization effects compared to oxadiazole as a spacer (see Figure 24).

## 6. Quinazolinone Hybrids

Several quinazolinone derivatives have been synthesized to provide synthetic drugs as more effective medicines for several disorders [69,70,71]. Quinazolinone hybrids constitute a crucial class of compounds with diverse therapeutic properties due to various substitutions on the ring system. Importantly, some quinazolinone analogs exhibited excellent anticancer activity via inhibition of dihydrofolate reductase enzyme [72]. Quinazolinones have also served as anti-tumor agents and inhibitors of tubulin polymerization [73]. Zayed and co-workers reported new hybrid molecules containing quinazolinone nucleus merged with various forms of L-amino acids at position three by exploring the structure of erlotinib **84** and thymitaq **85** [74]. The compounds were exposed to cytotoxic screening using two breast cancer cell lines, mainly MCF-7 and MDA-MB-231. All the synthesized compounds showed good activity against the MCF-7 cell line. Compound **86a** displays better activity than the reference erlotinib against the MCF-7 cell line. The most active compound, **86a** (*IC*_50_ = 0.44 ± 0.01 µM), against MCF-7, contains an L-phenylalanine substitution at position 3 of the quinazolinone scaffold. In comparison, **86b** (*IC*_50_ = 0.43 ± 0.02 µM), the most active compound against the MDA-MBA-231 cell line, has L-glutamine at position 3 of quinazolinone. Compound **86c** was the least active derivative for both cell lines. The tubulin polymerization assay revealed **86b** as a potent tubulin inhibitor (*IC*_50_ = 6.24 µM), which may explain these compounds’ high cytotoxic activity. The in silico study detailed that **86b** occupied the same landscape in the colchicine binding site with the formation of a strong hydrogen bond between the compounds and tubulin residues Tyr*α*224, GlnA111, a Gln*β*247, and Leu*β*248, respectively (see Figure 25).

## 7. Quinolinone Hybrids

Heterocycle-fused quinolinone is another privileged structure used as a building block in drug discovery. The quinolinone hybrids have shown biological activities, including antimicrobial [75], anti-inflammatory [76], anticancer [77], anti-Alzheimer [78], and anti-leishmanial [79] effects. Interestingly, quinolinone derivatives can successfully inhibit IDH1 mutants R132H, R132C, R132G, and R132L and display good selectivity vs. the wild-type IDH protein [80]. In addition, 7-phenyl-pyrroloquinolinones (7-PPyQs) and, mainly, the 3-substituted derivatives bind to the colchicine site with high affinity. Obstructing microtubule assembly and thus engendering excellent antiproliferative effect. This makes the compound have similar inhibitory activities to CA-4. It was believed that the [3,2-f] configuration, the 7-phenyl, and 9-carbonyl groups, and the ethyl group at position 3 would be crucial for suitable anti-tubulin activities. Considering these facts, Bortolzzi and colleagues designed synthesized hybrids mimicking PPyQ [81]. Compounds **88a** and **88c** emerged as the most active against a panel of seven human tumor cell lines including human T-cell leukemia (Jurkat and CEM), human B-cell leukemia (RS4; 11), human myeloid leukemia (Kasumi-1), breast (MDA-MB-231), cervix (HeLa), lung (A549) and colon (HT-29) cell lines as reported in the work of Bortolzzi [81]. The replacement of the ethyl group of **87** with a bulky linear alkyl chain (18 C) such as (R = CH_2_)_17_CH_3_, R_1_ = CH_3_) and (R = CH_2_)_17_CH_3_, R_1_ = phenyl) caused a drastic loss of antiproliferative activity in all cell lines. The SAR evaluation indicated that the increase in the alkyl chain attached to 3*N* decreased cytotoxicity. Compounds **89**, **90**, and **91** displayed moderate to poor activity, while **92**, a diazepine-indole derivative, displayed poor activity (GI50s > 10,000 nM). Moreover, **88a**, **88b**, and **88c** inhibited tubulin assembly with *IC*_50_ values of 0.99, 1.1, and 0.84 µM, respectively, which was higher than the *IC*_50_ values of the reference compounds (CA-4, *IC*_50_ = 0.64 µM). Notably, the anti-tubulin activity corroborated the growth inhibitory effects of the tested compounds. At position seven, the fused ring and the substituents established hydrophobic interactions with the residues at the *β*-subunit of tubulin dimers. Additionally, **88c** induced mitochondrial depolarization and ROS production, PARP activation, and decreased the expression of anti-apoptotic proteins (see Figure 26, Table 6).

## 8. Chalcone Hybrids

Chalcones are open-chain compounds with a chemical framework of 1,3-diaryl-2-propen-1-one, also known as chalconoid, which occurs as *trans* and *cis* isomers, with two aromatic rings linked to three-carbon *α*,*β*-unsaturated carbonyl. The thermodynamical stability of the *trans* isomer has made it the leading configuration among the chalcones. In contrast, the configuration of the *cis* isomer is thermodynamically unpredictable due to the sturdy steric influences between the carbonyl group and one of the aromatic rings. Chalcone derivatives and their various biological importance have been extensively studied, and several literature reviews have been published on chalcone derivatives [82,83,84,85,86,87]. Recent literature has highlighted certain chalcones’ anti-proliferative and tumor reduction activities, causing a resurgence of interest in this class of molecules.

In the eagerness of synergism in terms of activity, with enhanced affinity and efficacy about the parent structural components, Sultana and colleagues designed some chalcone conjugates by replacing the benzene ring of the chalcone **94** with benzo[d]imidazo[2,1-b]thiazole in the YM-201627 **93** structural compound [88]. Meanwhile, **93** has been reported as an effective and orally active antitumor agent [89]. Various substituents on both the pharmacophores were explored, and 27 compounds were synthesized and evaluated for biological activities by the authors. In addition, four panels of human cancer cell lines, mainly lung (A-549), breast (MDA MB-231), prostate (DU-145), and colon cancer (HT-29), were used in the MTT assay. All the compounds exhibited moderate to good antiproliferative activity with *IC*_50_ values ranging from 1.28 to 50 µM. Among the derivatives, the most active conjugates were **95a** (R = OCH_3_, R_1_ = R_4_ = H, R_2_ = OH, R_3_ = OCH_3_) and **95b** (R = R_1_= R_4_ = R_2_ = H, R_3_ = OCH_3_) with *IC*_50_ values 1.3 and 1.2 µM against MDA MB-231. The order of potency in regard to the substituents on ring D is 4-OCH_3_ > 3-OH- 4- OCH_3_ > 3, 4-diOCH_3_ > 3, 4, 5-triOCH_3_ > 2-Br-3,4,5-tri OCH_3_ > 3, 5-diOCH_3._ The anti-tubulin evaluation of the two compounds showed better *IC*_50_ values (1.93 and 1.88 µM) than nocodazole (*IC*_50_ =1.97 µM). The potent compounds further arrest MDA MB-231 cells in S and G2/M phase, disrupt the microtubule dynamics, produce ROS, and induce apoptosis. Meanwhile, molecular docking showed that the carbonyl oxygen group in **95a** and **95b** formed H-bonds with amide protons of Ala250*β* and Asn249*β* amino acid residues (see Figure 27).

In the continuous search for novel tubulin polymerization inhibitors using molecular hybridization strategy, Wang et al. [90] combined the pharmacophoric moiety of millepachine **96** and naphthalene **97**. The compounds were subjected to in vitro anticancer screening against HCT116 (human colon carcinoma cells) and HepG2 (human hepatocellular liver carcinoma cells) using the MTT method and millepachine as the reference compound. In contrast, when the electron-withdrawing group was inserted at the ortho or para position, this decreased anti-cancer activity. Moreover, an increase in inhibitory activity was observed when a methoxy substituent was positioned at the para position of the phenyl ring. However, the addition of more methoxy groups decreased the inhibitory activity. Compound **98a** exhibited excellent anti-cancer activity with an *IC*_50_ value of 1.20 ± 0.07 and 1.02 ± 0.04 µM against HCT116 and HepG2 cell lines. Compared to the reference millepachine (*IC*_50_ = 4.66 ± 0.23 and 1.51 ± 0.03). Compound **98a** proved to be a potent inhibitor of tubulin polymerization, with an *IC*_50_ of 22 µM as compared to standard colchicine (*IC*_50_ = 9 µM) and arrests HepG2 cells at the G2/M phase in a dose-dependent manner (see Figure 28).

Another publication has reported the synthesis and evaluation of aminochalcone hybrids as potential anti-cancer agents by targeting tubulin colchicine binding site [91]. Aminochalcones **98a** was fused with 3-amino-4-methoxyphenyl **99** moiety to form a single entity **100**. The structural modification of **100** generates another derivative **101**. The compound exhibited low cytotoxicity on normal human cell lines (LO2). In addition, compound **100** exhibited the most potent inhibitory activity against HepG2 (liver) and HCT116 (colon) human cancer cell lines with *IC*_50_ values of 0.28 ± 0.06 and 0.19 ± 0.04 µM, respectively, and inhibited tubulin polymerization (*IC*_50_ = 7.1 µM). It possesses the ability to arrest the cell cycle at the G2/M phase (see Figure 29).

Similar studies reported substitutions of the 2′-methoxy in the chalcone ring of HMNC-74 **102** with a hydroxy-functional group, and the synthesized compounds were evaluated for their anti-cancer activity [92]. The compound **103a** bearing 3-hydroxyl-4-methoxy phenyl moiety was observed as the most potent compound with an *IC*_50_ value of 1.42 ± 0.15 µM among the tested compounds compared to the reference compounds cisplatin (*IC*_50_ = 15.22 ± 1.27) against the MCF-7 (breast) cancer cell line. This promising compound **103a** was proven to have reduced cytotoxicity on the normal cell line (HEK293) compared to the tumor cell line. The activity of the tested compounds was affected by the nature of the substituents. For example, the substitution of the halogen group (R = 4-Cl or 4-Br) at the para position of the phenyl ring resulted in a slight increase in the anti-cancer activity. However, changing these groups to the meta or ortho position reduced the activity. While the replacement of the 4-Br or 4-Cl substitutes with the dialkylamine group also decreases the inhibitory activity, the insertion of the thiophene ring on the phenyl ring decreases antiproliferative activity. The most active compound (**103a**, *IC*_50_ = 8.4 µM) was slightly superior to colchicine (*IC*_50_ = 10.6 µM) in their evaluation for tubulin polymerization inhibition. The compound binds to tubulin and inhibits microtubule polymerization. In addition, compound **103a** arrests cells in the G2/M phase and stops cell mitosis, leading to the inhibition of MCF-7 cells proliferation. The adoption of an “L-shaped” conformation in the active site of tubulin for compound **103a** was revealed by molecular docking. The 2-methoxynaphthyl group was deep in the hydrophobic pocket containing residues Cys241, Leu248, Ala250, Leu252, and Leu255, whereas residue Lys254 formed a cation–π interaction with the phenyl group (centered ring) (see Figure 30).

Considering the tubulin polymerization activities of reported compounds bearing chalcone, trimethoxyphenyl, and diaryl ether, for example, 104, 20, and 105, [93,94,95], Wang et al. synthesized chalcone–diaryl ether hybrids (**106**). Compound **106a** with 4-methoxy substitution on the chalcone ring, which was found to show superior inhibitory activity on MCF-7(breast), HepG2 (liver), and HCT116 (colon) cancer cell lines, with *IC*_50_ values of 3.44 ± 0.19, 4.64 ± 0.23, and 6.31 ± 0.27 μM, respectively [96]. The active compound accelerates cell cycle progression and induces cell apoptosis through tubulin polymerization inhibition (104a, *IC*_50_ = 20.0 μM and colchicine, *IC*_50_ = 10.6 µM). Detailed molecular docking studies involving **106a** and tubulin showed that Asn101 and Ser178 interact with **106a** via hydrogen bond formation (see Figure 31).

Based on the unrelenting effort of these authors to search for anti-cancer agents, they reported for the first time the synthesis of chalcone hybrids containing naphthalene **107** and indole **108** moieties [97]. Position five was maintained for *α*,*β*-unsaturated ketone group for derivations because shifting it to another position displayed slightly to moderate anticancer activity **109**. Out of the 18 synthesized compounds, only **110a** has excellent cytotoxic activity against HepG2 (liver), HCT116 (colon), and MCF-7 (breast) with *IC*_50_ values of 0.65, 1.13, and 0.82 μM, respectively. Meanwhile, substituting benzyl or bulk alkyl groups at the *N*-1 position of the indole ring enhances biological activity. The authors further investigated the anti-tubulin activity of the active compound. The result showed that it inhibited tubulin polymerization with *IC*_50_ of 3.9 µM (compared to colchicine, *IC*_50_ = 13 µM) and caused cell cycle arrest in G2/M in a dose-dependent manner (see Figure 32).

A series of resveratrol–cinnamoyl hybrids **114** as tubulin polymerization inhibitors were synthesized and showed moderate antiproliferative activities towards four-panel cancer lines [98]. The authors introduced the acyl ester group **113** for better interaction of the compounds with the targeted protein. Compound **114a** exhibited the most potent antiproliferative activity with *IC*_50_ values of 0.12, 0.016, 0.44, 0.37, and 0.78 μΜ against A549, MCF-7, HepG2, HeLa, and MDA-231, respectively. This activity was superior to the reference drug colchicine. The SAR showed that the electron-donating group (R = OCH_3_) on the benzene ring of chalcone improved the antiproliferative activity and the electron-withdrawing groups (R = NO_2_, or CF_3_) decreased the inhibitory activities of the compounds. Compound **114a** was observed to inhibit tubulin polymerization (*IC*_50_ value = 1.03 μM) marginally better than CA-4 (*IC*_50_ = 1.32 μM). The visualization of the molecular docking depicted that the **114a** and colchicine occupied similar binding landscapes. At the same time, the 3,4,5-trimethoxyphenyl of **114a** formed a hydrogen bond with the critical amino acid Cys241, which anchors the majority of CBS inhibitors. In addition, the presence acyl ester group enhanced the flexibility of the compound, which allows **114a** to bind to the colchicine site of tubulin with excellent conformation. The potent compound disrupts the microtubule dynamics, and the binding domain landscape was found to be similar to that of colchicine (see Figure 33).

The analogs of chalcone bearing shikonin moiety (**51**) were synthesized and screened against three cancer cell lines, namely HeLa (cervical), MCF-7 (breast), and A549 (lung), as well as two non-cancer cell lines: 293T (epithelial) and L02 (hepatic) under in vitro condition by Qiu et al. [99]. Compound **51a** with an OCF_3_ substituent was found to be the most potent with *IC*_50_ values of 4.53 ± 0.07 2.36 ± 0.32 and 5.84 ± 0.18 µM, respectively, against HeLa, MCF-7, and A549 cancer cell lines, and exhibited a substantial anti-tubulin effect with an *IC*_50_ value of 2.98 ± 0.53 µM. The mechanism of action of **118a** was observed to be like colchicine, which includes inducing apoptosis in MCF-7 cells lessening mitochondrial membrane potential, as well as arresting the cell cycle at the G2/M phase, and the disorganization of the microtubule system. (see Figure 34)

Based on the panorama, the 3,4,5-trimethoxyphenyl (TMP) moiety of CA-4 is responsible for its strong activity because of the formation of a hydrogen bond between the 4-methoxyl group and residue Cys241 [100]. Xu and colleagues expected that the *N*-1 atom of pyridine would also form a hydrogen bond with residue Cys241 in the colchicine binding sit. They replaced the trimethoxyphenyl ring of **104** with the pyridine **119** scaffold. A series of novel pyridine-chalcone (**120**–**121**) derivatives as potential anti-tubulin agents have been reported [101]. Compound **121a** exhibited the most potent activity with the *IC*_50_ values ranging from 0.023–0.047 µM against four panels of cancer cell lines and one normal cell line. Compound **121a** (R_1_ = R_2_ = R_4_ = OCH_3_, R_3_ = OH, and R_5_ = H) demonstrated an *IC*_50_ value of 0.023 µM against K562 cell lines which was ~3-fold superior to compound **104**. The methyl-substituted at the *α*-position of an unsaturated carbonyl group in **120, 121** series were favored for their inhibitory activity. Meanwhile, the hydroxyl on ring B was crucial for maintaining the inhibitory activity of **121** hybrids. In addition, the substitution of electron-donating groups on the pyridine (**121**) enhanced the activity compared to electron-withdrawing groups. The authors reported that electron-withdrawing groups improved the electron density of the *N*-1 of pyridine, thus allowing the interaction of the compound with residue Cys241 through H-bonding as they expected. The 3-substituted indoles exhibited good inhibitory activity (**124, 125**), while substituents at *N*-1 of the indole ring showed low inhibitory activity against K562 (**126**–**128**). Compound **121a** inhibited superior tubulin polymerization with an *IC*_50_ value of 2.08 µM, compared to CA-4 (IC50 = 2.17 µM). The potent compound further caused G2/M phase arrest, induced cell apoptosis, and disintegrated the intracellular microtubule assembly. Compound **121a** positively inhibited the tumor volume and lessen the tumor weight (65.8% at 20 mg/kg per day (i.v.)), compared to the reference compound CA-4 **1** (50.9% at 20 mg/kg) or CA-4P (62.7% at 20 mg/kg). The active compound also acts as a vascular disrupting agent (see Figure 35).

The broad-spectrum activity of sarcodictyin A **129a**, and sarcodictyin B **129b**, a natural compound with anti-tubulin activity, have impelled many medicinal chemists to explore the total synthesis and subsequently derivatization. In addition, the *α*, *β*–unsaturated hetero-aromatic structure and the imidazole ring are both responsible for the compound’s strong activity thus the substitution of imidazole with another heterocycle group will disfavor the activity of the compounds. Hence, Oskuei et al. [102] designed and synthesized a new series of imidazole-chalcone conjugates **130** as anti-tubulin and anti-cancer agents. The compounds were evaluated against four panels of cancer lines, namely, MCF7 (breast), A549 (lung), HepG2 (liver), and MCF7/MX (breast) by MTT assay. Compounds **130a** and **130c** exhibited excellent antiproliferative activity with *IC*_50_ values ranging from 7.05 to 63.43 μM. The substitution of Br atom at R_2_ strongly enhanced the cytotoxic activity against A549 cancer cells, comparable to compound **130a**, which can be caused by proper size and the lipophilicity of the bromine (Br). However, the presence of chlorine and fluoride at the same position reduced the compound’s effectiveness. The oxidation of **130b** (R_1_ = R_2_ = R_3_ = OCH_3_ and R_4_ = SCH_3_) led to the formation of **131** (R_1_ = R_2_ = R_3_ = OCH_3_, and R_4_ = SO_2_CH_3_), enhanced the cytotoxicity of the compound on MCF-7. Both **130a** (42.85% and 60.71%) and **130c** (25% and 55.71%) inhibited tubulin polymerization at concentrations of 50 and 100 μM, respectively. In comparison, the tubulin inhibitory activity of compound **130b** was inferior to the inhibitory activity of **130a** and **130c**, which corroborated their antiproliferative activities. The potent compounds further arrest the cell cycle at the G2/M phase at low concentrations and increase apoptotic cells (cells in subG1 phase) at higher concentrations. The molecular docking showed that **130a** was well fitted in the hydrophobic pocket and interacted with residues, Glu183*α*, Thr224*α*, Lys254*β*, Asn101*α*, Val351*β*, Lys352*β*, and Leu 248*β*, respectively. In addition, **130a**, H-bonded with the catalytically active residues Ser178*α* and Ala316*β* and accompanied with a cation–π interaction (Asn258*β*) (see Figure 36, Table 7).

## 9. Coumarin Hybrids

Coumarin is also known as 2*H*-1-benzopyran-2-one, benzo-*α*-pyrone, phenylpropanoids, cis-*o*-coumarinic acid lactone, coumarinic anhydride, and 5,6-benzo-2-pyrone. They are secondary metabolites of plants, bacteria, fungi, and sponges. Physically, coumarin is a crystalline colorless solid with a specific sweet odor. Coumarin has been classified as an oxygenated heterocycle rather than a benzoic acid derivative. Many coumarin hybrids have been reported to be used for medical disorders, namely as anti-allergic [103], antiviral [104], vasodilatory [105], antibacterial [106,107], exhibiting cyclooxygenase inhibition [108], lipoxygenase [109], antithrombotic [110], and anticancer activity [111]. The molecular hybridization of the coumarin nucleus to a biodynamic system that is a biologically active pharmacophore has enhanced biological activities. These compounds with heterocycle-coumarin’s broad pharmacological activity have gained momentous attention in medicinal chemistry. Coumarin has also been reported to possess many pharmacological activities, such as tubulin polymerization inhibition. In the view of coumarin **132** and cyanohydrazone **133** as cytotoxic agents, Govindaiah and colleagues reported for the first time the amalgamation and cytotoxicity activity of 4′,7-dihydroxycoumarin bearing cyanohydrazone moieties [112]. A series of 4,7-dihydroxycoumarin based acryloylcyanohydrazone hybrids **135** was synthesized and evaluated for antiproliferative activity against a panel of four cancer cell lines, namely A549, HeLa, SKNSH (neuroblastoma), and MCF7. Among the tested compounds, **135a**–**135d** showed notable antiproliferative activity with *IC*_50_ values ranging from 3.42 to 10.26 μM against all the tested cancer cell lines. **135c** (R = 4-N(CH_3_)_2_) emerged as the most active compound with IC50 values of 4.31 ± 0.04, 5.14 ± 0.16, 6.09 ± 0.32, and 3.42 ± 0.52 μM against A549, HeLa, SKNSH, and MCF7, respectively. The SAR detailed that substitution at the para position (R = 4-OCH_3_, 4-Br, 4-N(CH_3_)_2_ and 4-NO_2_) and ortho position compounds (R = 2-Cl and 4-NO_2_) increases the anticancer activity compared to substitution at the meta position compounds (3-NO_2_ and 3-Br). The substitution of the electron-donating group at the para position enhanced the inhibitory activity more than the electron-withdrawing group. The authors further evaluated the pharmacological mechanistic studies of **135c** on cell cycle progression and carried out a tubulin polymerization inhibition assay. The results indicate that **135c** arrests the cell cycle at the G2/M phase and inhibits tubulin polymerization with an *IC*_50_ value of 6.19 μM; thus, the compound can be a candidate antimitotic agent for treating cancer by targeting tubulin protein. The docking studies revealed that hydrazide–hydrazone backbone is more active when the cyano group is attached to the hydrazone moiety: This improved the activity of the compound (see Figure 37).

In the continuation of their research, they further synthesized novel hydroxycoumarin-based cyanohydrazone compounds **137** and screened for antiproliferative activity against A-549, HeLa, SKNSH (neuroblastoma), MCF-7 human cancer cell lines, and a normal rat kidney cell line (NRK-49F) using the MTT assay. All the screened compounds exhibited excellent antiproliferative activity with *IC*_50_ values ranging from 8.20 to 27.39 μM. Among the evaluated compounds, **137a** was observed as the most potent compound with *IC*_50_ values of 8.64 ± 0.13, 8.36 ± 0.12, 9.26 ± 0.04, 8.20 ± 0.18, and 33.42 ± 0.06 µM against A-549, Hela, SKNSH, MCF-7, and NRK-49F, respectively. The pharmacological studies of **137a** revealed that it inhibited MCF-7 cells proliferation through cell cycle arrest at the G2/M phase, followed by cell death, and significantly inhibited tubulin polymerization with an *IC*_50_ value of 11.03 μM, which is comparable to that of the standard drug colchicine (*IC*_50_ = 7.75 μM) (see Figure 37).

Singh and co-workers [113] synthesized and evaluated triazole tethered isatin–coumarin bifunctional hybrids (**140**). The hybrids were tested against a panel of four human cancer cell lines using the sulforhodamine B assay. As the electronegativity substituent was inserted into the isatin moiety, the compounds (**140b**, **140c**, and **140d**) showed superior activity; meanwhile, the activity decreased pointedly with the increase in chain length of the spacer (**140d**, **140e**). The remarkable decrease in cytotoxicity may probably be due to an increase in the size of the molecule, which may not be perfectly fitted within the active binding site of tubulin. The most active compounds (**140a**, **140b**, and **140d**) were investigated further for their inhibitory effects on tubulin polymerization using a standard tubulin polymerization assay kit. Superior anti-tubulin activity was observed in compound **140a** with an *IC*_50_ value of 1.06 µM, whereas **140b** and **140d** inhibited tubulin polymerization with *IC*_50_ values of 3.55 and 6.32 µM; hence, these three compounds exhibited their cytotoxic effect through tubulin inhibition (see Figure 38, Table 8).

## 10. Indole Hybrids

Indole is also known as benzopyrrole, a bicyclic heterocyclic molecule with a benzenoid system. It has ten π-electrons, lone pairs from nitrogen, and four double bonds that provide eight valence electrons. When an electrophilic substitution occurs at C-3 of indole moiety, it becomes more reactive than benzene. Indole scaffolds have the exceptional characteristics of imitating the structure of peptides and binding in a reversible way to enzymes that offer enormous possibilities for discovering new medicines with different modes of action [114]. Many natural, semi-synthetic, and synthetic hit molecules containing indole scaffolding have been described in the literature and are either commercialized or in various stages of drug development. In addition, several tubulin polymerization inhibitors bearing indole rings have been obtained from natural sources or synthesized. The mechanistic anti-tumor activity of vincristine and vinblastine is through their effects on tubulin polymerization. Vincristine is an anti-tumor agent well-known as a tubulin polymerization inhibitor and used in combination therapy to treat acute lymphoblastic leukemia, Hodgkin’s, and non-Hodgkin lymphoma [115,116]. Vinblastine is primarily used to treat advanced Hodgkin’s disease, breast cancer, Kaposi’s sarcoma, renal cell cancer, and testicular cancer [117,118,119,120,121,122,123]. These compounds are M phase cell cycle-specific, prevent the growth of the mitotic spindle, induce a terminal mitotic phase, and cause cell death.

Many indole-based compounds have been reported as tubulin polymerization inhibitors binding to the colchicine domain, including IPP51 (**141**) and JAI-51 (**142**). IPP51 is selective toward the proliferation of bladder carcinoma cells and vies with colchicine for binding to the tubulin [124]. While the trimethoxy analog **142** showed antiproliferative activity against four human and a murine glioblastoma cell line, interacted with tubulin, and fitted into the colchicine binding site [125]. Meanwhile, MX58151 (**143**) and crolibulin (**144**), bearing a 3-bromo-4,5- dimethoxy-phenyl moiety, are new microtubule inhibitors [126,127]. Based on the usefulness of 3-bromo-4,5-dimethoxy-group and indole moiety in antiproliferative and tubulin polymerization inhibitory effectiveness, Mirzaei and colleagues combined chalcone and indole scaffolds into a single entity, which are indole-based chalconoids [128]. The amalgamation of indole-chalcone scaffolds (**145**) and in vitro biological evaluation as tubulin-targeting anti-cancer agents were reported. The in vitro antiproliferative activity of the synthesized compounds were examined against adenocarcinoma human alveolar basal epithelial cells (A549), breast cancer cells (MCF7), and human ovarian carcinoma cells (SKOV3) using an MTT assay. *(E*)-stereoisomer was assigned for all the synthesized compounds because the detected coupling constants (*J*) between vinylic hydrogens were more significant (*J* = 15 Hz). Among the synthesized compounds, **145a** (*IC*_50_ value = 4.3 ± 0.2 µM) showed superior cytotoxicity against human alveolar basal epithelial (A549) and SKOV3 cells, while **145b** was more potent against the MCF7 cell line (11.7 ± 0.2 µM). In addition, **145a**, **145b**, and **145c** inhibited tubulin polymerization (*IC*_50_ values = 17.8 ± 0.2, 18.3 ± 0.4, and 40.0 ± 0.1 µM, respectively). Compound **145a** reduced the mitochondrial thiol content in a concentration-dependent manner. Cytotoxic effects of **145a** can be possibly correlated to tubulin polymerization inhibition, reactivity toward mitochondrial thiols, and induction of mitochondrial pathway of apoptosis. The methoxy groups on the phenyl ring and the carboxyl group of the *α*, *β*-unsaturated ketone moiety are deepened into the pockets of the colchicine binding site as detailed in the molecular docking studies. The *α*, *β*-unsaturated ketone forms hydrophobic interaction with Ala250 and Lys254 residues, while the bromine atom on the phenyl ring interacted with Ala317 lys352 residues in chain B. (see Figure 39).

Substituting the double bond in CA-4 (**1**) with the hybridized carbonyl group(sp2) provides molecules such as phenstatin **146**, which will maintain the cis orientation of the two rings, a feature of CA-4, liable for presenting better biological properties [129,130]. Many phenstatin-containing molecules have been reported without removing the 3,4,5-timethoxyphenyl ring but substituting the 3-hydroxy-4-methoxy phenyl ring with various heterocyclic rings, including the BPR0L075 (**149**)-bearing carbonyl bridge and indolyl scaffold. BPR0L075 can effectively impede cancer cell growth and tubulin polymerization [131,132]. These observations prompted Kode and colleagues to report alterations and the biological effectiveness of phenstatin, indole, and chalcone scaffolds. Subsequently, a new phenstatin based indole-linked chalcone compounds (**151**) was designed, synthesized, and evaluated against several cancer cell lines [133]. Compound **151a** (R_1_ = CH_3_, R_2_ = R_3_ = OCH_3_, and R_4_ = H) demonstrates highly significant inhibitory activity against SCC-29B oral cancer cells with GI_50_ < 0.1 µM. In contrast, high inhibitory activity against HEP-G2 cells was observed for compound **151b** (GI_50_ = 0.018 µM), and compound **151c** (GI_50_ = < 0.1 µM) showed significant inhibitory activity against HT-29 colon cancer cells. Compound **151a** was further evaluated for its effect on tubulin polymerization. The results demonstrated that it causes destabilization of tubulin, leading to the disruption of microtubules. It also caused the loss of a microtubule-organizing center (MTOC) closer to the nucleus and degradation of cell integrity. The SAR of the synthesized compounds showed that the presence of trimethoxy groups and the carbonyl group enhanced the tubulin activity and the placement of methoxy substituents at R_2_ and R_3_ improved cytotoxicity (GI_50_ value = < 0.1µM). The presence of trimethoxy ring and carbonyl group critical for tubulin activity enhanced cytotoxic activity. The compound interacted with the target binding site and inhibited microtubule polymerization and glucose metabolism. Compound **151a** successfully reduced tumor volume and angiogenesis in mice. It also lessens collagen levels and causes an obvious reduction in cell processes, cellular integrity, and cytoskeletal organization (see Figure 40)

A series of new curcumin-indole analogs (**155**, **156**), with indole, and phenyl moieties linked on either side of the 1,5-diaryl-1,4-pentadien-3-one system, were synthesized. The compounds were screened for their cytotoxic potential against a panel of eight cancer cell lines, namely, lung (A549), breast (MDA-MB-231, BT549, and 4T1), prostate (PC-3, DU145), gastric (HGC-27), and cervical (HeLa) by Ramya and co-workers [134]. Compounds featuring N-benzylated indole analogues (155) exhibited lesser activity than N-unsubstituted indole analogs (**156**). Notably, among all the compounds tested, compounds **156a**, **156b**, and **156c** were significantly most active on HGC-27 with *IC*_50_ values of 5.21, 7.45, and 8.65 µM, HeLa (*IC*_50_ of 6.59, 6.44, and 3.31 µM, respectively) and BT549 cell lines (*IC*_50_ of 8.34, 4.69 and 8.72 µM respectively). Compounds **156b** and **156c** showed the most potent antiproliferative activity on PC-3 cell lines with *IC*_50_ values of 6.34 ± 0.25 and 3.15 ± 1.91 µM. Additionally, compound (**156b**) was tested on RWPE-1 (normal prostate) cells and superior to PC-3 cells. The SAR analysis showed that substituting electron-donating groups (OCH_3_) enhanced the antiproliferative activity while substituting the electron-withdrawing group (4-Cl) reduced the inhibitory activity. The authors further examined the potent compounds to investigate whether they interact with the microtubule system, prompting antiproliferative activity. The results showed that **156a** and **156c** powerfully inhibited microtubule assembly with *IC*_50_ values of 10.21 ± 0.10 and 8.83 ± 0.06 µM, respectively (see Figure 41).

The combination of nicotinic acid and pyrazoline into a single scaffold enriches the molecular stability and prevents the oxidative property of the pyrazoline scaffold, whereas the methylation of the nitrogen atom of indole significantly enhances the potency of tubulin polymerization inhibition and cytotoxicity [135]. The molecular hybridization of these compounds has been reported as potential tubulin polymerization inhibitors [136]. Based on previous research on nicotinic acid and pyrazoline, Chen and colleagues [137] envisage that the amalgamation of indole, nicotinic acid, and pyrazoline will improve tubulin inhibitory activity. Hence, twenty-five nicotinoyl pyrazoline derivatives bearing *N*-methyl indole were synthesized (**161**). The authors evaluated their antiproliferative and tubulin polymerization inhibitory properties against four human cancer cell lines. The cancer cell lines include MCF-7 (human breast adenocarcinoma), A549 (human non-small cell lung carcinoma), HepG2 (human liver hepatocellular carcinoma), and HeLa (human cervix carcinoma), respectively. Notably, some of the compounds showed potent antiproliferative activity, and compound **161f** displayed the most potent activity against MCF-7, A549, HepG2, and HeLa cells with GI_50_ values of 0.09, 0.59 0.029, and 0.034 µM, respectively. The order of inhibitory activity was as follows for different substitutions: 3-OCH_3_ > 4-OCH_3_ > 4-F > 4-Cl > 4-Br, when R_1_ was substituted with methoxy, and R_3_ was substituted with 6-methoxy. Meanwhile, among the compounds tested for tubulin assembly inhibition **161a**–**161g** exhibited excellent inhibitory activity comparable to the standard compound. In addition, **161f** showed superior anti-tubulin assembly with an *IC*_50_ value of 1.6 µM, compared to the reference drug CA-4 (*IC*_50_ value = 2.1 µM). The molecular docking result of **161f** showed that the 3,4,5-trimethoxy phenyl interacted with residue Cys*β*241 via H-bond, while meta-methoxy phenyl formed an H-bond interaction with Cys*β*241 in the hydrophobic pocket containing residues Leu*β*248, Leu*β*255, Ala*β*316, Ala*β*317, Val*β*318, Ala*β*354, and Leu*β*378, respectively. In the in vivo experimentation of **161f** on HeLa-Xenograft nude mice, the relative tumor ratio was up to 61.52% with insignificant weight loss and tissue damage, comparable to CA-4 (see Figure 42, Table 9).

Considering the therapeutic importance of indolyl-3-glyoxylamides and 3-aminoindole scaffold, Diao et al. [138] designed indole-based oxalamide and aminoacetamide by integrating glyoxylamide and acetamide moiety into the 3-aminoindole nucleus (**162, 163**). The compounds were synthesized and examined for their antiproliferative activities (**164**–**165**). The indole derivatives bearing oxalamide (**164**) group exhibited inferior inhibitory activity (*IC*_50_ > 100 μM), while aminoacetamide derivatives showed greater antitumor activities. More importantly, compounds **165a**–**165d**, which incorporate aminoacetamide moiety at position three of the indole ring, exhibited potent antiproliferative activities with *IC*_50_ values of 16.20 ± 0.63, 18.30 ± 1.30, 16.74 ± 2.45, 15.41 ± 1.80 µM, respectively, against the HeLa cancer cell line. The substitution of phenyl ring with pyridine nucleus leads to slightly enhanced antiproliferative activities. Notably, compounds **165a**–**165d** exhibited anti-tubulin activity (7, 29, 43, and 46%, respectively, at a concentration of 10 µM). Compound **165d** with superior inhibition of tubulin polymerization disrupts the microtubule organization in the HeLa cells. Moreover, molecular docking analysis indicated that **165d** formed balanced interactions in the colchicine binding site in the *α*/*β*-tubulin interface between tubulin subunits (see Figure 43).

The indole-1,3,4-oxadiazole (**167**) and indole-1,2,4-triazole (**168**) hybrids were synthesized in two series by replacing the imidazole scaffold of topsentin^122^. The synthesized compounds were further evaluated for their antiproliferative activity using 60 human cancer cell lines panels. The 1,3,4-oxadiazole containing derivatives displayed superior activity compared to 1,2,4-triazole derivatives. Among the compounds tested, compounds **167a**, **168a**, and **169b** proved to have excellent inhibitory activity with *IC*_50_ values of 2.42, 3.06, and 3.30 μM, respectively, against the MCF-7 human cancer cell line compared to the reference drug doxorubicin (*IC*_50_ value = 6.31 μM). Of all cancer cell lines, the colon (HCC-2998) and breast (MCF-7, T-47D) cancer cell lines were found to be more vulnerable to this class of compounds. The SAR analysis showed that aryl substitution on the rings enhanced superior antiproliferative activity compared to the aliphatic substitution. Moreover, the substitution with aliphatic groups in the heterocyclic moieties resulted in activity loss. Electron-withdrawing groups on aromatic rings displayed better activity than electron-donating groups, and the activity patterns of the substituent were found as follows: F > Br > NO_2_ > Cl. Further, **167a** and **168a** were tested for tubulin polymerization inhibitory activity, and the results showed that **167a** and **168a** disrupt microtubule dynamics by inhibiting tubulin polymerization with *IC*_50_ values of 3.89 and 8.32 μM, respectively. In addition, **167a**, which appeared as a lead molecule, induced cell cycle arrest at the G0/G1 phase and disrupted mitochondrial membrane by reducing the cell migration potential of MCF-7 cells in a dose-dependent manner compared with standard nocodazole (*IC*_50_ 2.49 μM) (see Figure 44).

Reports have shown the anti-tubulin potency of coscinamides (**169**), hyrtiosin B (**171**) which are natural containing compounds [139,140,141]. In addition, indibulin (**170**) bearing 1,2-diketo or *α*-keto amide functional group show strong anticancer activity [142]. Kumar et al. and Mukherjee et al. have demonstrated the anticancer activities of bis(indolyl)hydrazide-hydrazone (**172**) featuring hybrids [143,144]. Thus, compounds bearing pharmacophoric units such as *α*-ketoamide, diketone and hydrazide-hydrazone have therapeutic importance. Based on this panorama, Tantak and coworkers designed novel bis(indolyl)ketohydrazide-hydrazones by integrating *α*-ketoamide, diketone and hydrazide-hydrazone as potent tubulin inhibitors [145]. Twenty bis(indolyl)ketohydrazide-hydrazones conjugates were synthesized. They evaluated their anticancer activity against five-panel cancer cell lines by using ketohydrazid hydrazones as spacers for the two-indole ring in coscinamide moiety. The panel of cancer cell lines used in their studies includes human breast cell lines (BT474, MCF-7, and MDA-MB-231), human colon carcinoma cell lines (HCT-116), and leukemia cell line (Jurkat). The substitution of an indole ring in **173** with an aryl or pyridyl group resulted in compounds **174** with a percentage of cell survival more significant than 90% except for the Jurkat cancerous cell line. Compounds **173a**–**173e** displayed cytotoxic effects (<50% cell survival) against the cancer cell lines. **173d** emerged as the most potent anti-proliferative agent against MCF-7, MDA-MB-231, HCT-116, and JURKAT cancer cell lines with *IC*_50_ values of 0.8, 0.50, 0.15, and 0.22 µM, respectively. SAR studies indicated that C-5 methoxy, Bromo, and *N*-(4-chlorobenzyl) groups were necessary for the superior anti-cancer activity of ketohydrazide-hydrazones. The presence of bis-indole (**173**) rings compared to the aryl or pyridyl group (**174**) enhanced the inhibitory activity of ketohydrazide-hydrazones. Further, studies of the mechanism of action revealed that **82d** induces caspase-dependent apoptosis by activating caspase 3/7 but is non-toxic and stops the G2/M stage cell cycle by preventing the polymerization of the tubulin (*IC*_50_ = 0.6 µM) (see Figure 45).

## 11. Oxindole Hybrids

Many medicinal chemistry researchers have shown enormous interest in developing novel oxindole hybrids due to their invaluable biological potential. Marketed oxindole derivatives include sunitinib and indolidan. Sunitinib is used to treat gastrointestinal stromal tumors and advanced renal cell cancer [145], while indolidan is a phosphodiesterase inhibitor [146,147]. Other oxindole derivatives, such as SU5416, SU5402, and S 6668, have exhibited potent antitumor activity and are known as receptor tyrosine kinase (RTK) inhibitors [147,148]. 1,4-Dihydroindeno[1,2-c]pyrazole (**174**) has acted as an inhibitor such as cyclin-dependent kinase (CDK), PDGFR tyrosine kinase inhibitor, checkpoint kinase 1 (Chk1) inhibitors, tyrosine kinases (EGFR and VGEFR-2), hypoxia-inducible factor receptor (HIF-1) [149]. A novel series of 1, 4-dihydroindeno-[1,2-c] pyrazole in combination with the oxindole (**175**) scaffold have been synthesized by Khan and co-workers [149]. The compounds were synthesized using the Knoevenagel condensation approach and further evaluated for their antiproliferative activity against four panels of cancer cell lines, namely, HeLa, A549, and MDA-MB-231 human cancer cell lines as well as HEK-293 (normal human embryonic kidney cells). Among the tested compounds, **176a**–**176c**, superior cytotoxicity with *IC*_50_ values varying from 1.33 to 4.33 µM. Meanwhile, the tubulin polymerization assay demonstrated that **176a**–**176c** triggers microtubule assembly and boosts the G2/M checkpoint proteins (Cyclin B1 and CDK1), whereas compound **176a** induces apoptosis through the activation of caspase-3 besides. Importantly, the presence of a Cl or OCH_3_ substituent at ring E was found to be crucial for excellent anti-tubulin activity (see Figure 46).

The design, synthesis, and evaluation of a series of anilinopyridyl-oxindole hybrids (**178**) against a panel of four human cancer cell lines have been reported [150]. Interestingly, the evaluated compounds displayed good to moderate anti-cancer activity against the tested cell lines, namely, HeLa, DU-145, A549, MCF-7, and normal rat kidney cell lines (NRK-49F), respectively. Compounds **178a**–**178e** revealed to have potential antiproliferative activity with *IC*_50_ values of 1.3 ± 0.5, 1 ± 0.5, 0.9 ± 0.07, 1 ± 0.1, and 2 ± 0.4 µM, respectively, against DU-145. The placement of halides at position five of ring D enhanced the inhibitory activity compared to the substitution of halides at position six. The presence of methoxy (R = OCH_3_) on ring A reduced the inhibitory activity, while Cl, F, and CF_3_ groups enhanced the activity. The order of improved potency with the substituents was as follows; F > Cl > CF_3_ > H > OMe. Notably, substituting the electron-withdrawing group on rings A, D, and the electron-donating group on ring C enhanced the cytotoxic activity. The results of the tubulin polymerization assay showed *IC*_50_ values of 1.84 and 2.43 µM for **178c** and **178e**, respectively. The compounds inhibit microtubule assembly and disrupt microtubule structure in cells. The visualization provided by the related molecular docking studies demonstrated that the potent hybrids (**178c**, **178e**) snuggly fit into the colchicine binding site of the tubulin dimer and display both hydrophobic and hydrogen-bonding interactions with the closest residues, including, Val181*α*, Asn258*β*, Met259*β*, Thr314*β*, Val315*β*, Ala316*β*, Asn350*β*, Val351*β*, Lys352*β,* and Leu255*β* Val238*β*, Cys241*β*, Leu242*β*, Ala316*β*, Val318*β*, Leu248*β,* and Ala250*β*, respectively (see Figure 47).

## 12. Conclusions

The concept of designing hybrid molecules containing two or more chemical entities a single molecule is now used in drug discovery. Many medicinal chemistry researchers presently use the molecular hybridization approach in developing unique bioactive compounds acting as dual inhibitors, while being less susceptible to drug resistance. The molecular amalgamation involved the pharmacophoric unit of a known protypes [151] which are significantly active to produce a single entity. This approach was utilized (i) for modulation of objectionable secondary effects and (ii) to identify dual-acting molecule capable of reproducing the effect of combinations of more than one therapeutic agent. Triazole, tetrazole, benzimidazole, pyrazole, quinoline, quinazolinone, quinolinone, chalcone, coumarin, indole, and oxindole are attractive molecules with potent anticancer activities that have gained more attention thus they are used as building blocks by many researchers for designing and developing novel chemical structures. This review presented the recent literature related to the hybridization of heterocyclic organic molecules, as mentioned above, that exert their anticancer properties through disruption of microtubule structure in the cells. We also tried to shed light on structure–activity relations to the extent possible and provide an overview of compounds with moderate activities, which are too often ignored. Considering recent advancements described in this review article (2017–2021), it is evident that developing tubulin–olymerization hybrids of conventional compounds could augment their therapeutic effectiveness and overcome multi-drug resistance providing a promising path for more efficacious chemotherapy drug development less fraught with adverse side effects.

## Figures and Tables

**Figure 1 ijms-23-04001-f001:**
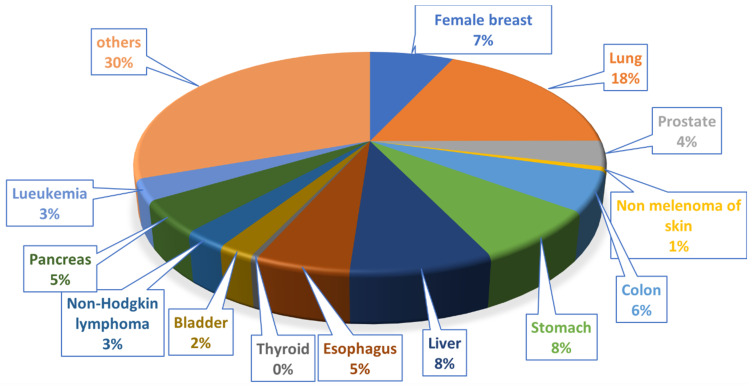
Global distribution of the top 13 cancer types with the highest mortality rate.

**Figure 2 ijms-23-04001-f002:**
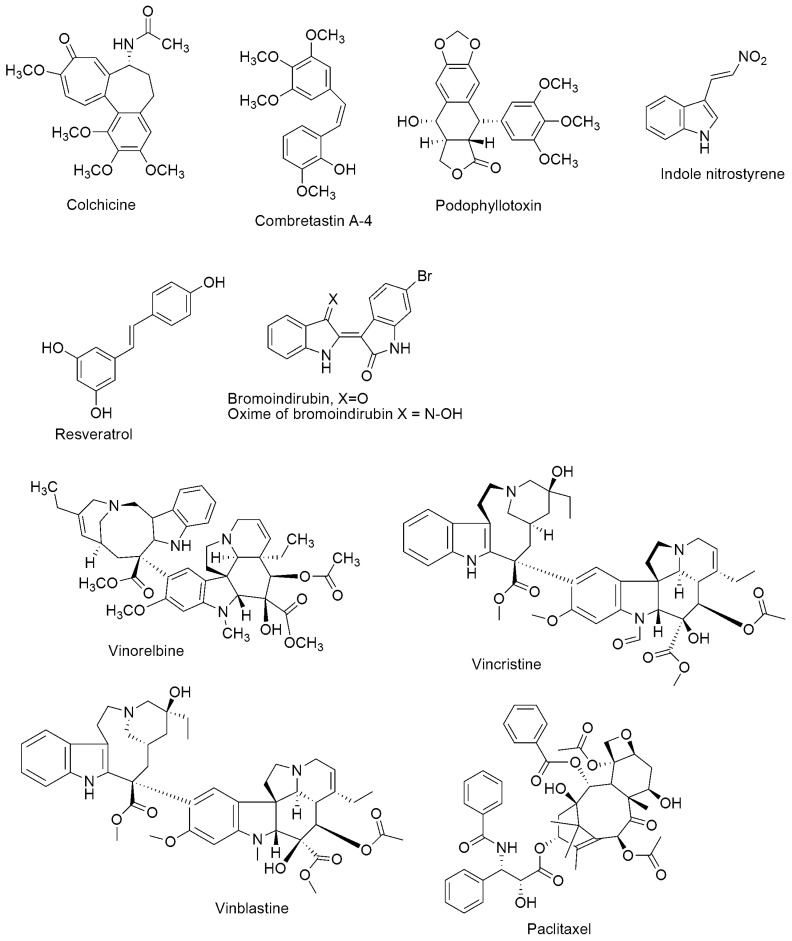
Chemical structures of tubulin polymerization inhibitors.

**Figure 3 ijms-23-04001-f003:**
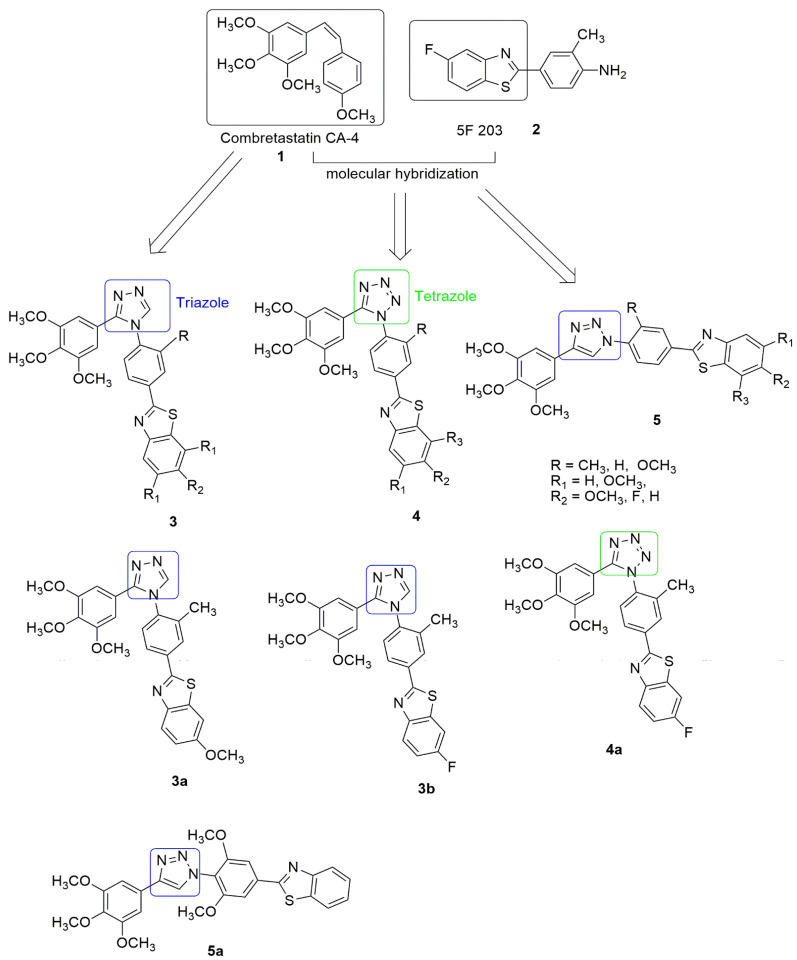
Chemical structures of triazole/tetrazole–benzothiazole hybrids with anti-tubulin activities.

**Figure 4 ijms-23-04001-f004:**
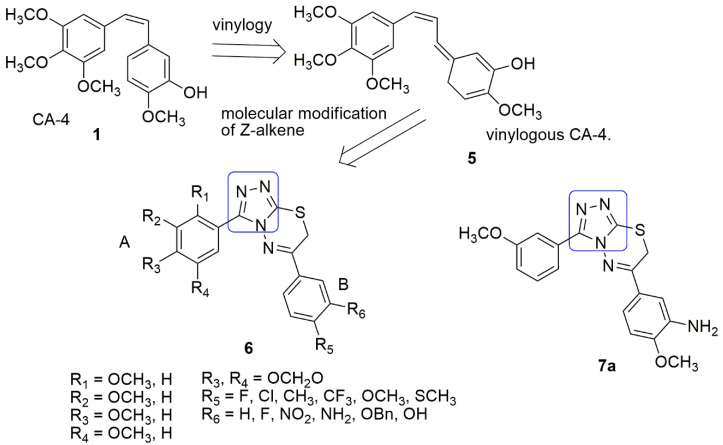
Chemical structures of triazole–thiadizine hybrids that inhibit tubulin polymerization.

**Figure 5 ijms-23-04001-f005:**
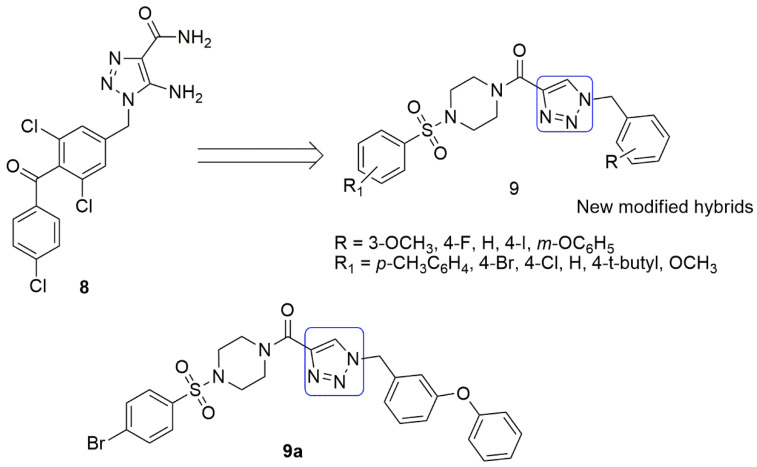
Chemical structures of 1,2,3-triazole, sulfonamide, and piperazine moieties that inhibit tubulin polymerization.

**Figure 6 ijms-23-04001-f006:**
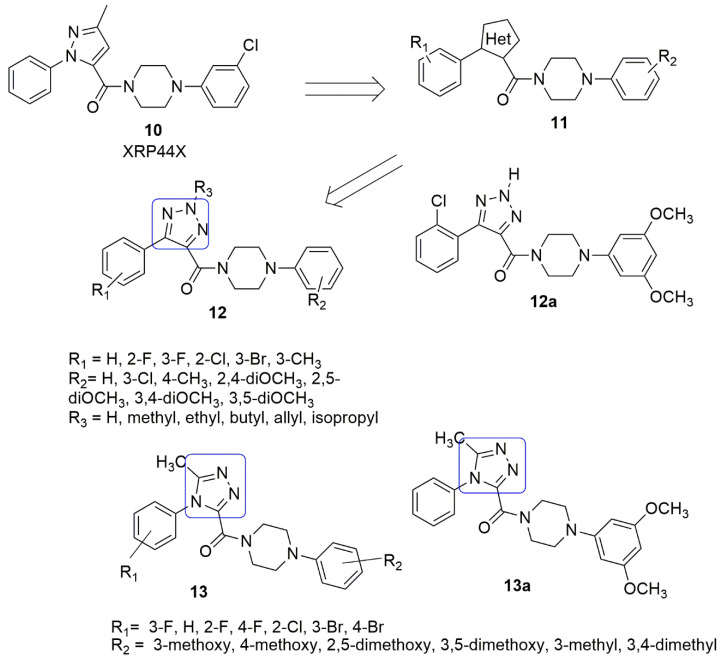
Chemical structures of arylpiperazine and triazole hybrids as a potential microtubule targeting agent.

**Figure 7 ijms-23-04001-f007:**
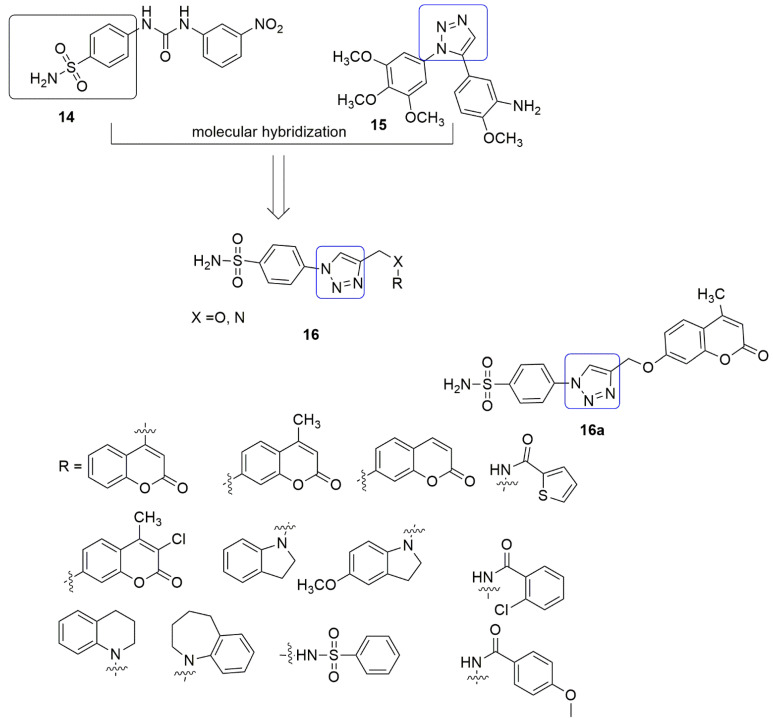
Chemical structures of sulfanilamide-1,2,3-triazole hybrids used against three selected human cancer cell lines (BGC-823, MGC-803, and SGC-7901).

**Figure 8 ijms-23-04001-f008:**
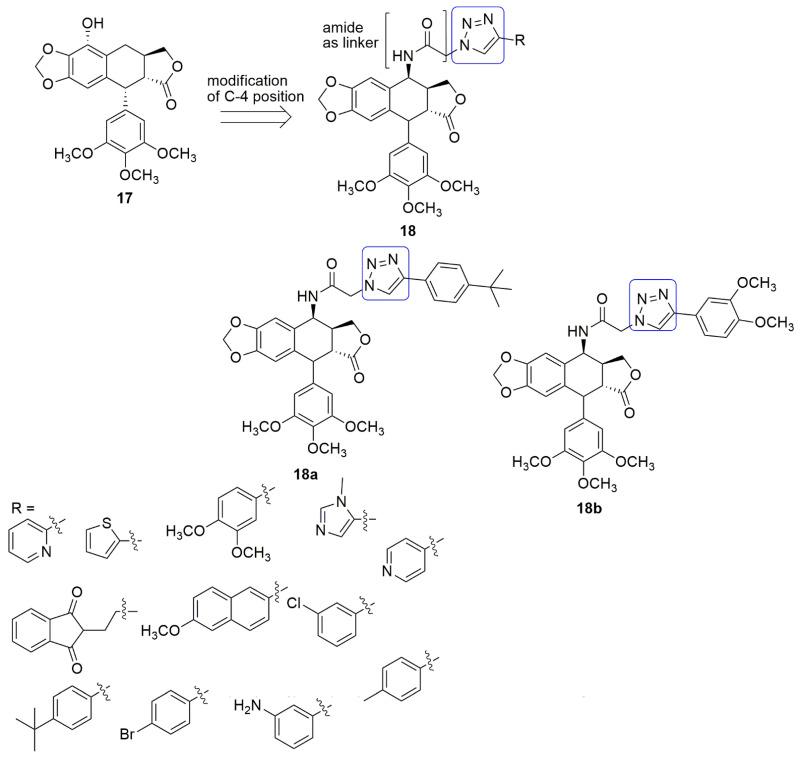
Chemical structures of triazolo-4*β*-amidopodophyllotoxin derivatives as promising anti-tubulin agents.

**Figure 9 ijms-23-04001-f009:**
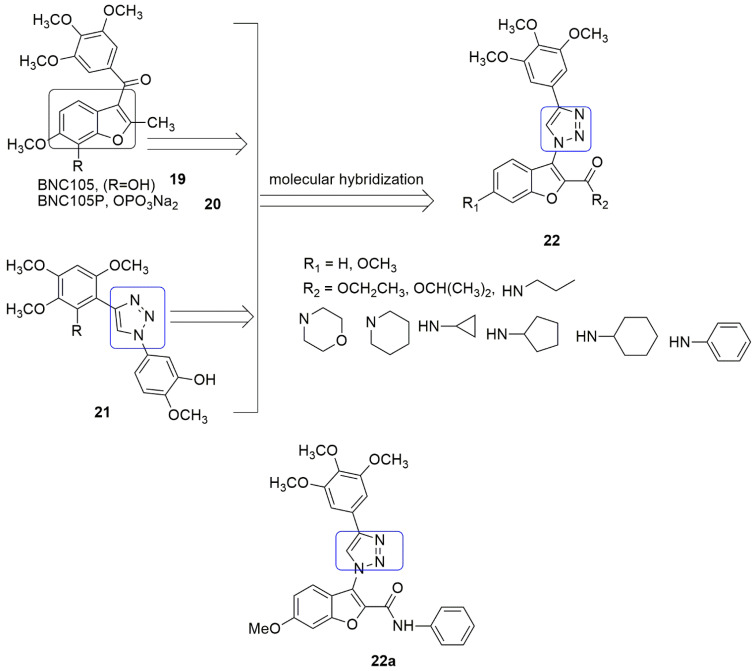
Chemical structures of benzo[*b*]furan and triazole moieties that displayed antiproliferative and anti-tubulin activity.

**Figure 10 ijms-23-04001-f010:**
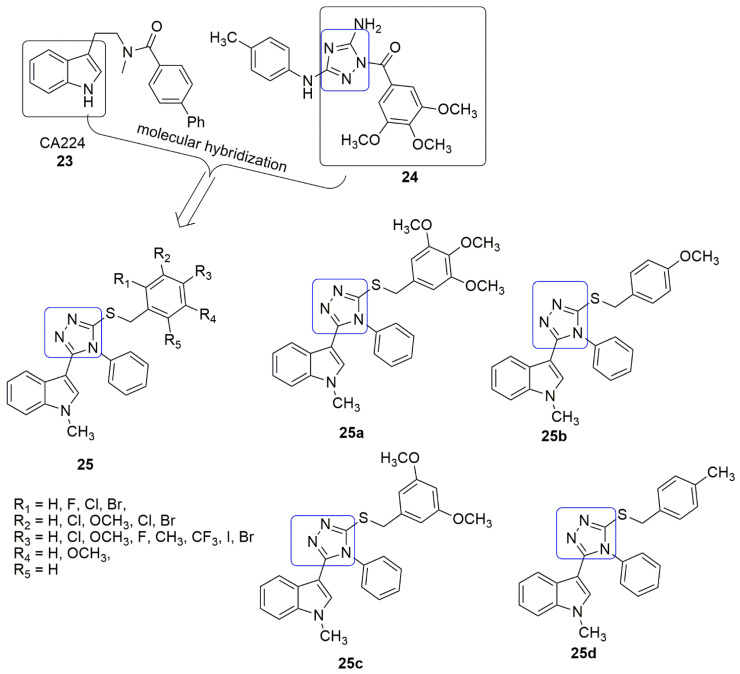
Chemical structures of indole-1,2,4-triazole scaffolds evaluated for anti-tubulin activity.

**Figure 11 ijms-23-04001-f011:**
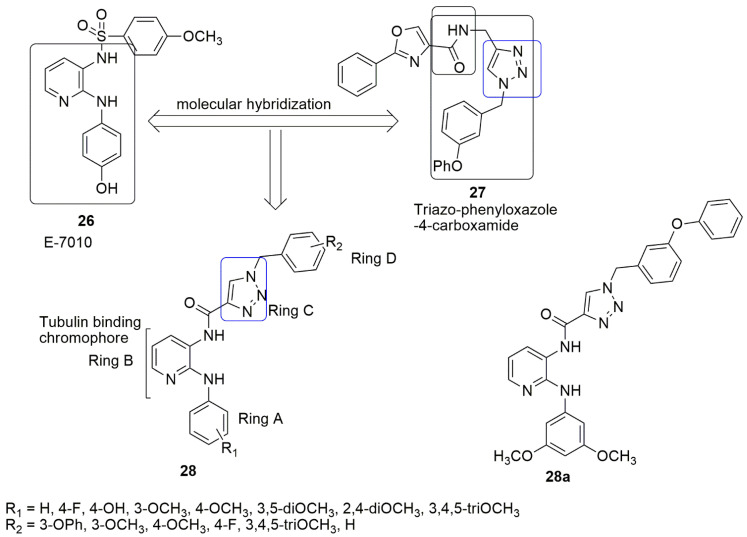
Chemical structures of pyridine-triazole carboxamide evaluated for cytotoxicity and tubulin polymerization.

**Figure 12 ijms-23-04001-f012:**
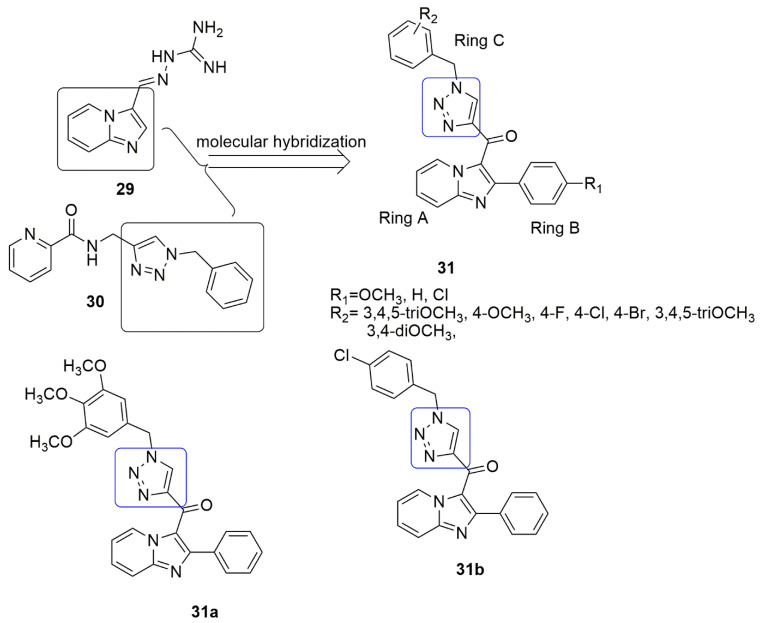
Chemical structures of triazole–imidazopyridine conjugates as promising anti-tubulin activity.

**Figure 13 ijms-23-04001-f013:**
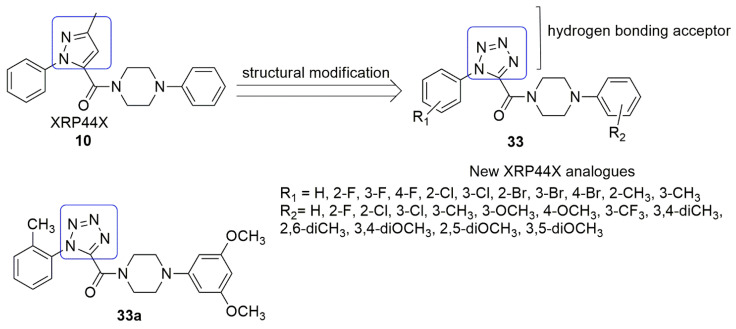
Chemical structures of 1-aryl-5-(4-arylpiperazine-1-carbonyl)-1*H*-tetrazole derivative as microtubule destabilizers.

**Figure 14 ijms-23-04001-f014:**
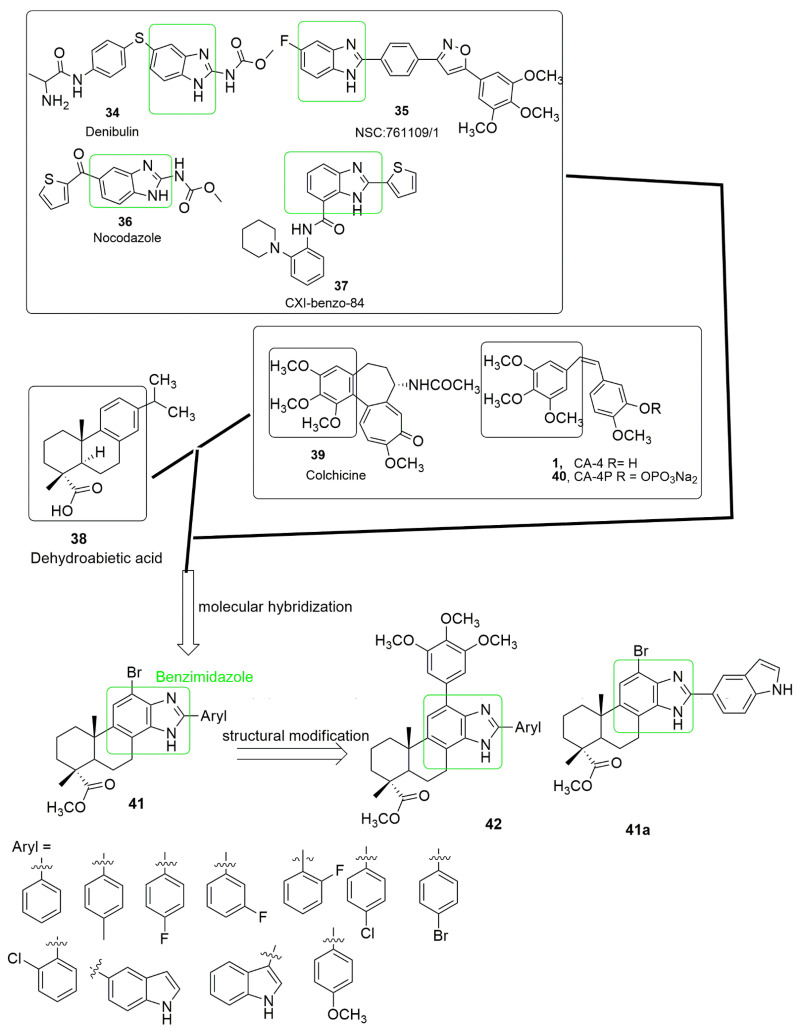
Chemical structures of dehydroabietic–benzimidazole hybrids.

**Figure 15 ijms-23-04001-f015:**
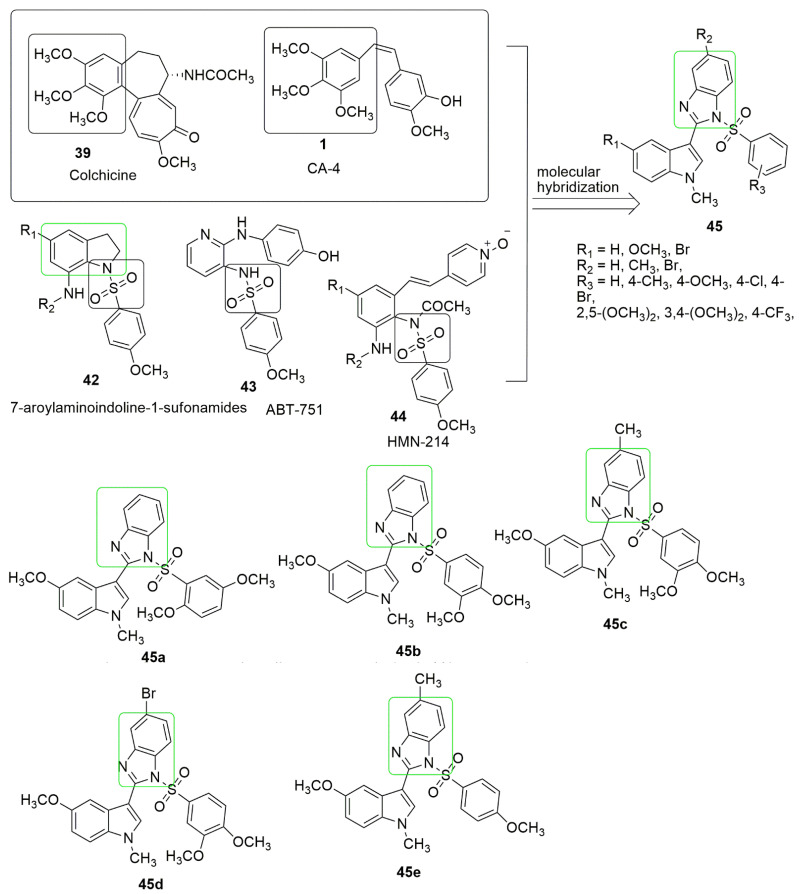
Active anti-cancer compounds of 1-phenylsulphonyl-2-(1-methylindol-3-yl)-benzimidazole derivatives.

**Figure 16 ijms-23-04001-f016:**
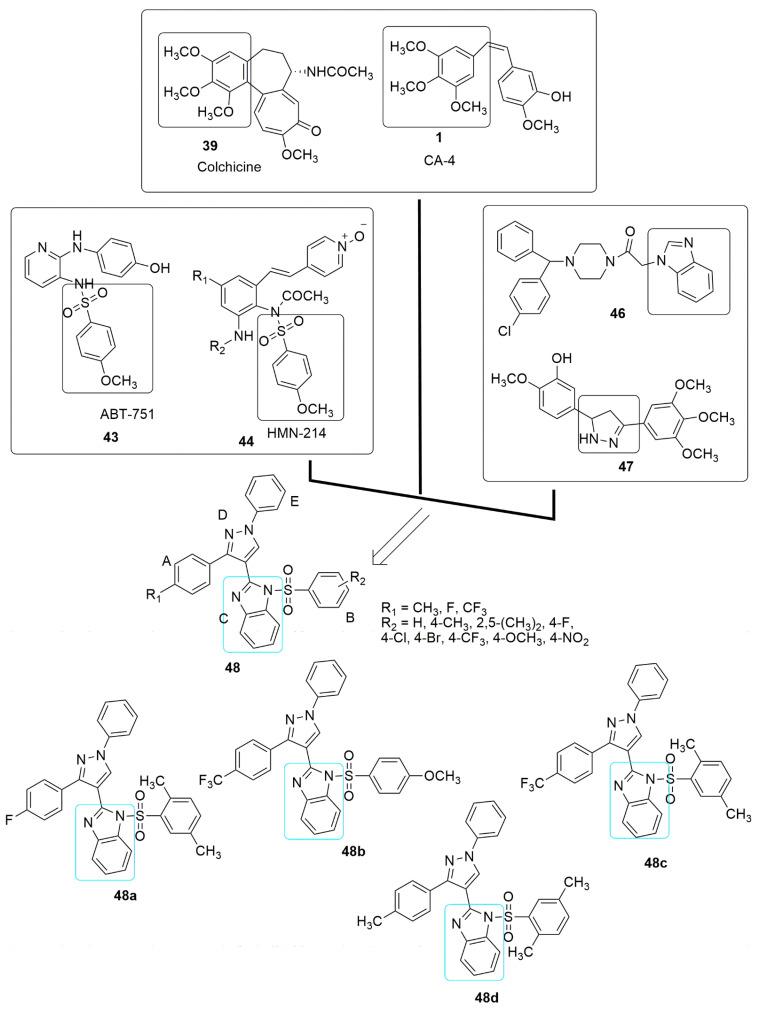
Antiproliferative and anti-tubulin activity of active compounds of benzimidazole conjugates.

**Figure 17 ijms-23-04001-f017:**
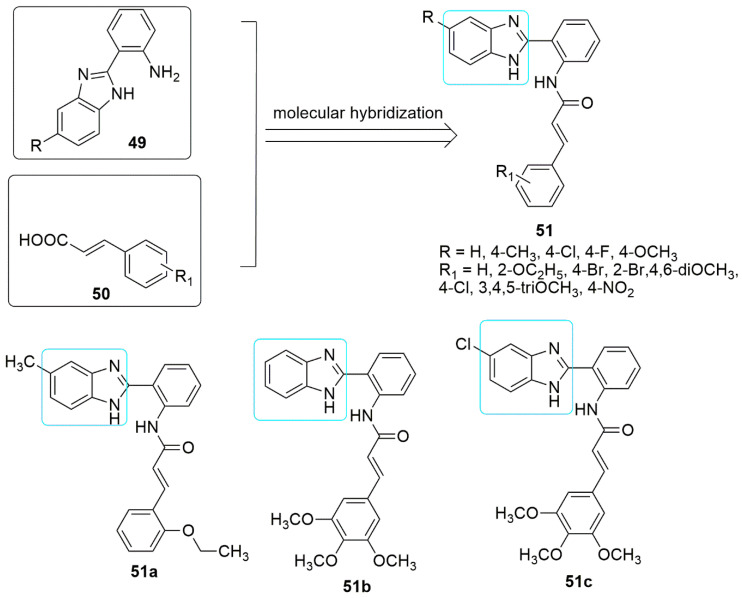
Chemical structures of benzimidazole and cinnamic acid hybrids.

**Figure 18 ijms-23-04001-f018:**
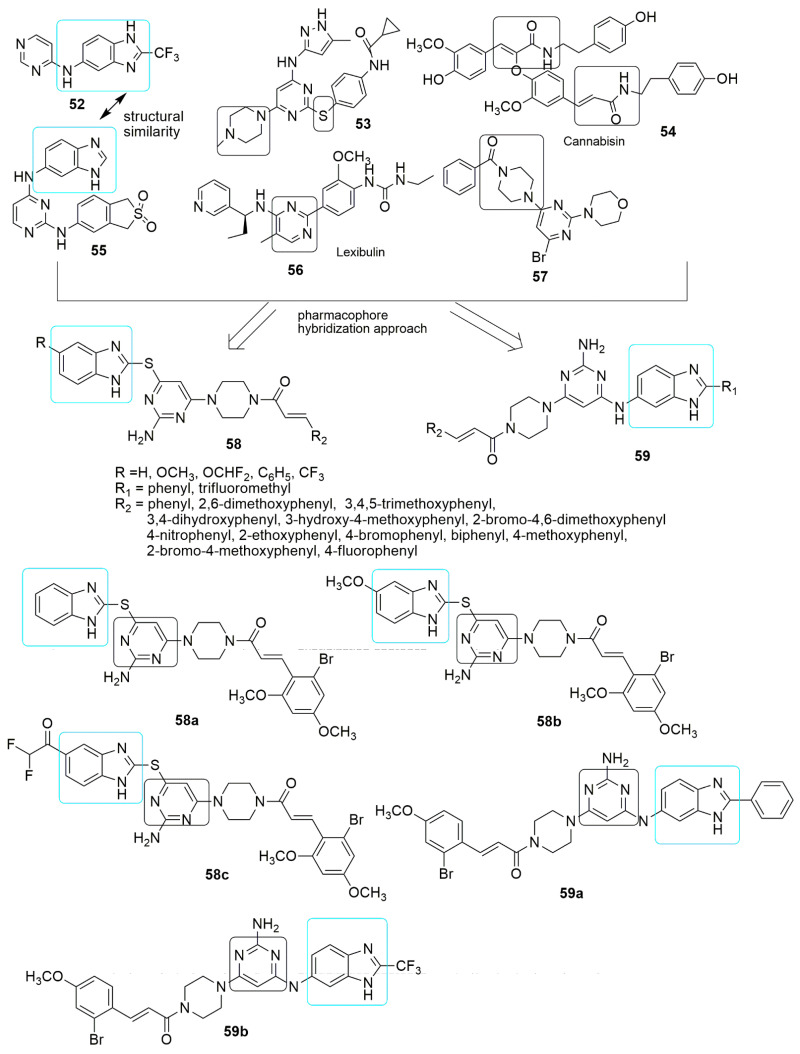
Pyrimidine-benzimidazole hybrids as anti-cancer agents.

**Figure 19 ijms-23-04001-f019:**
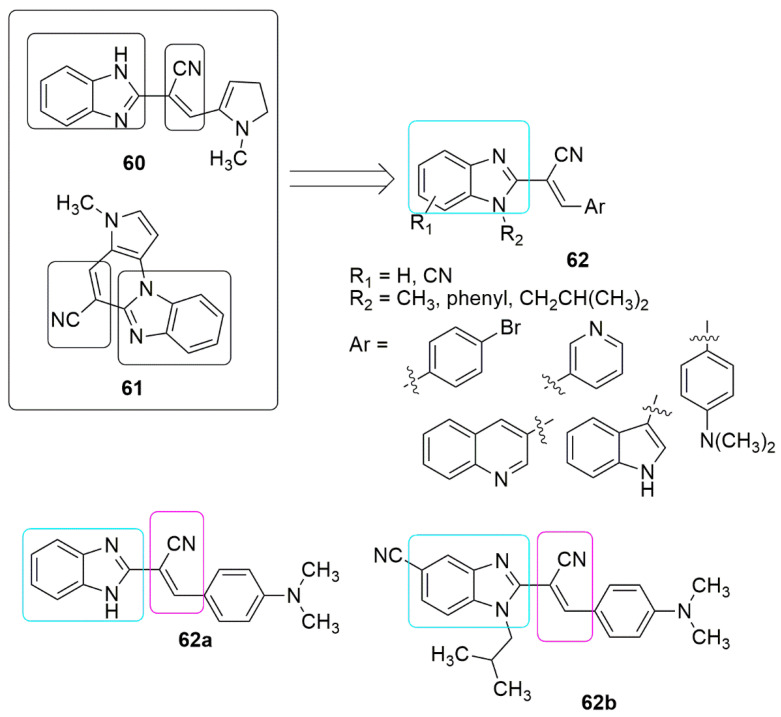
Chemical structures of *N*-substituted-2-benzimidazolyl acrylonitriles hybrids.

**Figure 20 ijms-23-04001-f020:**
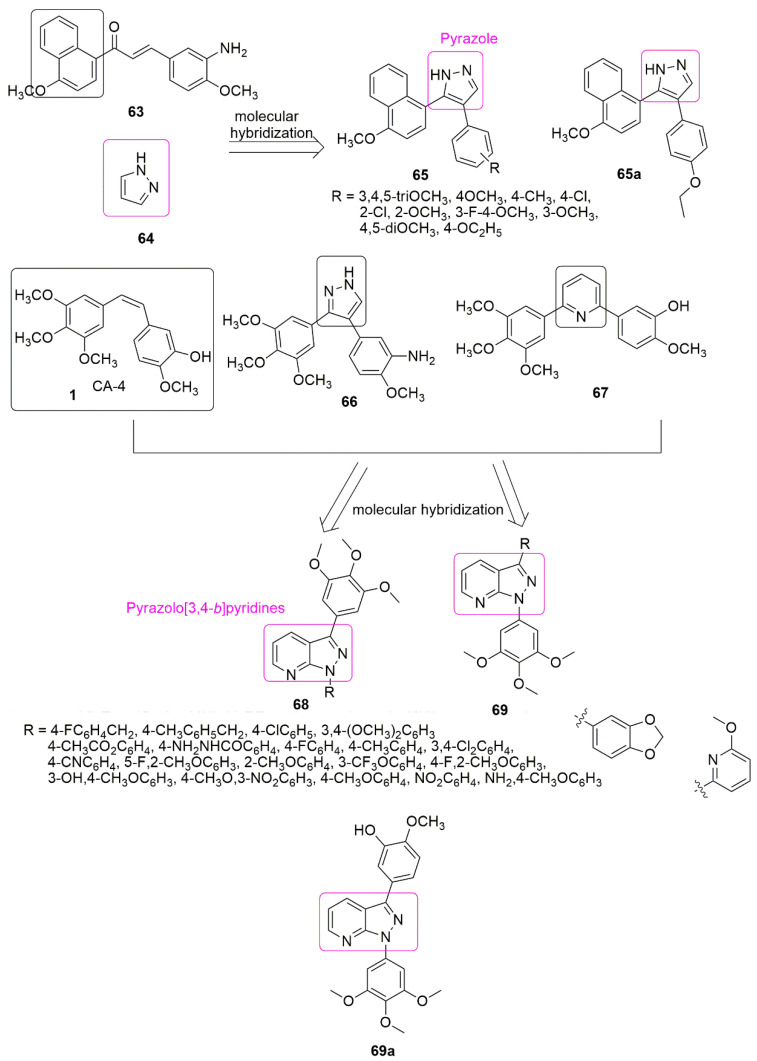
Chemical structures pyrazole hybrids inhibiting tubulin polymerization.

**Figure 21 ijms-23-04001-f021:**
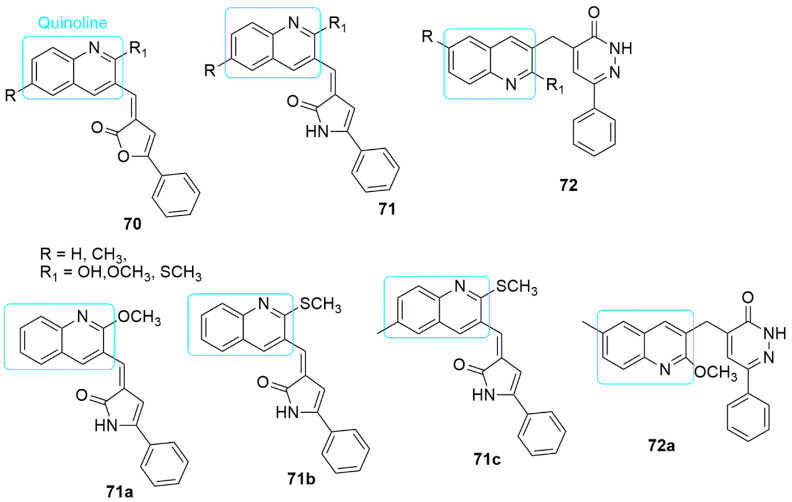
Chemical structures of quinoline conjugates as anti-tubulin agents.

**Figure 22 ijms-23-04001-f022:**
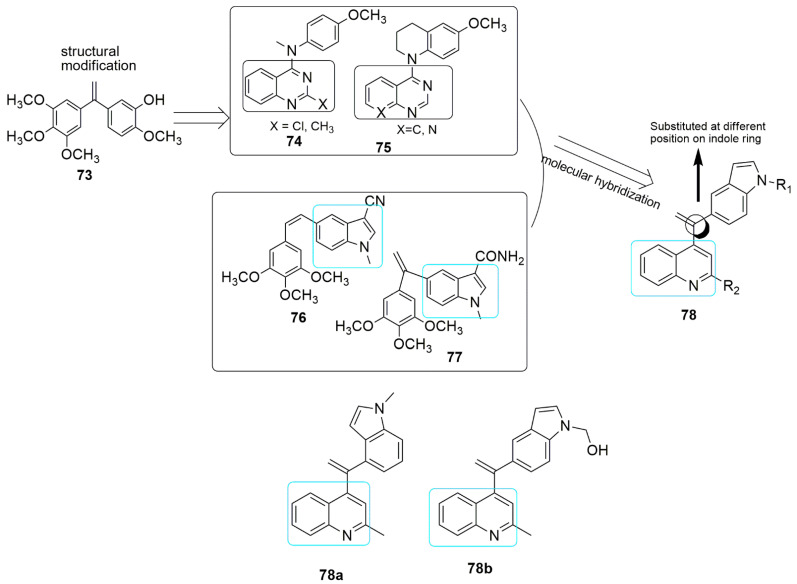
Chemical structures of quinoline evaluated against different panel of cancer cell lines.

**Figure 23 ijms-23-04001-f023:**
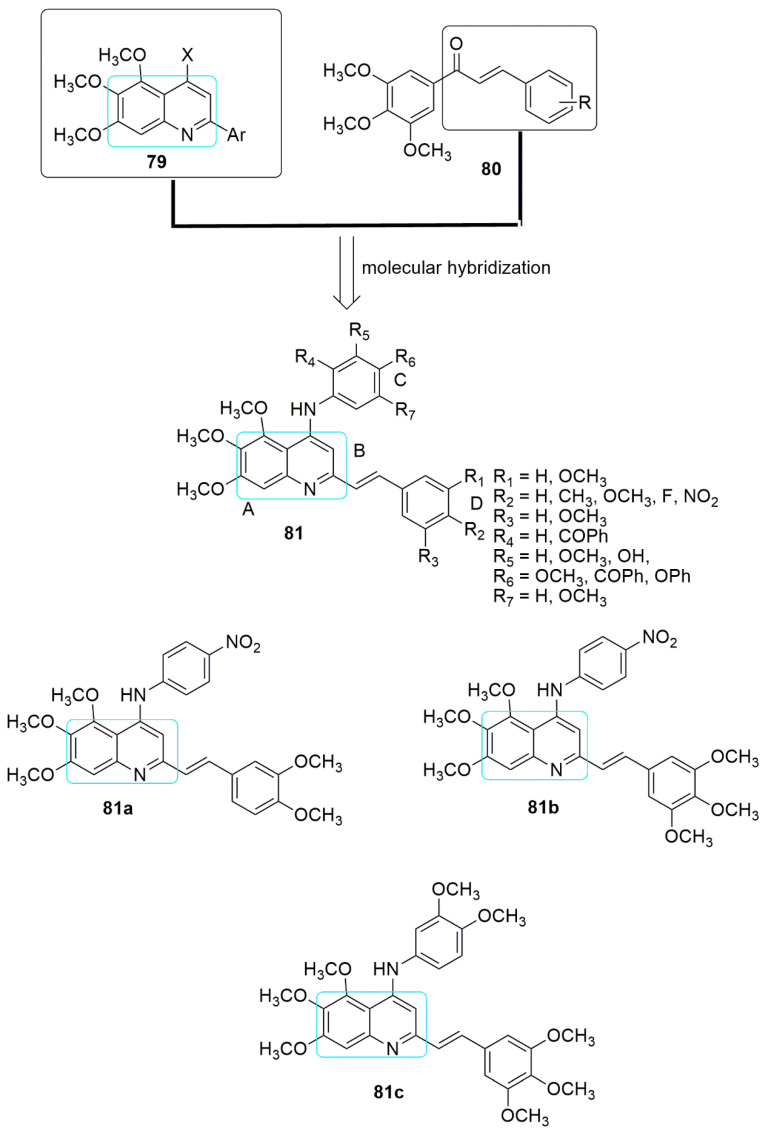
Promising anti-cancer agents and tubulin polymerization inhibitors.

**Figure 24 ijms-23-04001-f024:**
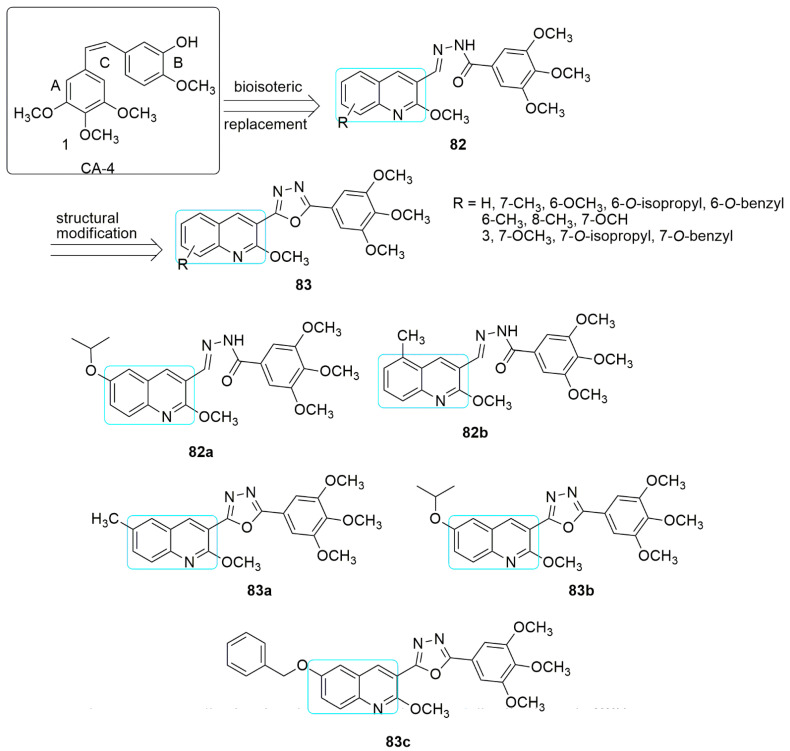
Chemical structures of quinoline conjugates of combretastatin A-4.

**Figure 25 ijms-23-04001-f025:**
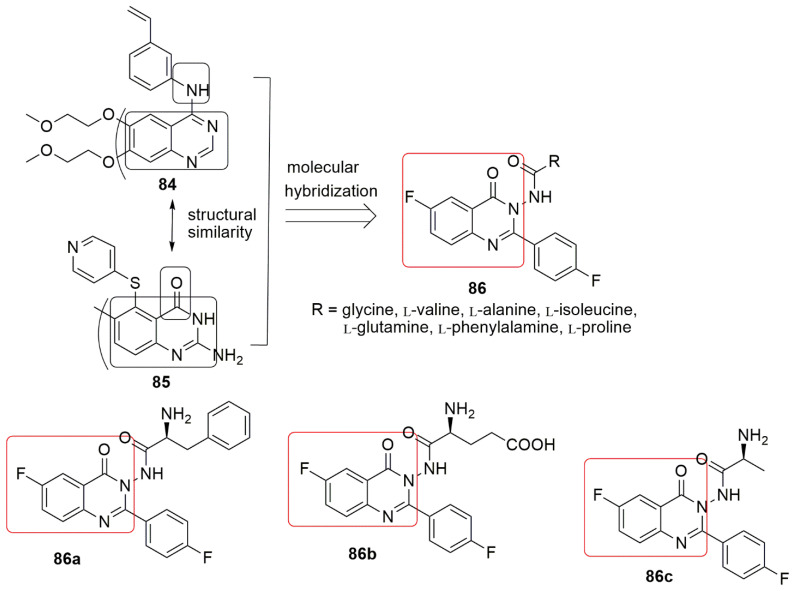
Chemical structures of quinazolinones derivatives evaluated for biological activity.

**Figure 26 ijms-23-04001-f026:**
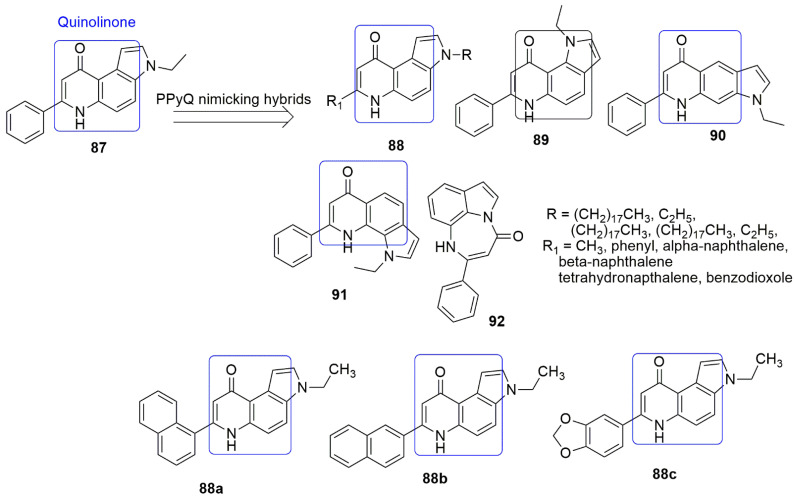
Chemical structures of quinolinone hybrids evaluated against a panel of cancer cell lines.

**Figure 27 ijms-23-04001-f027:**
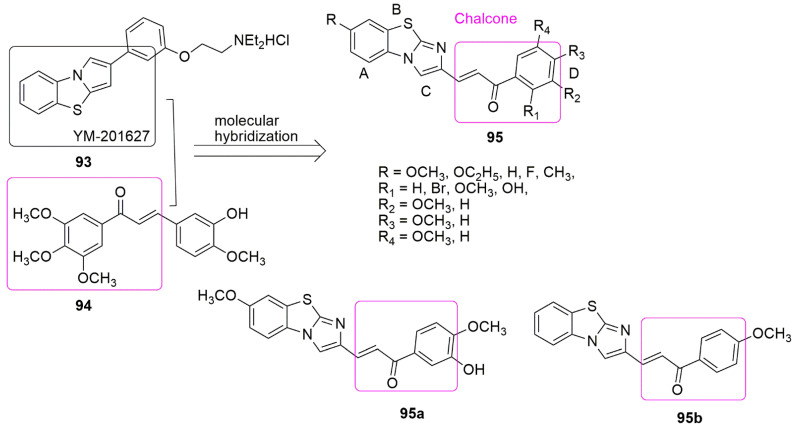
Chemical structures of benzo[*d*]imidazo[2,1-*b*]thiazole-chalcone conjugates as potent anti-tubulin agents.

**Figure 28 ijms-23-04001-f028:**
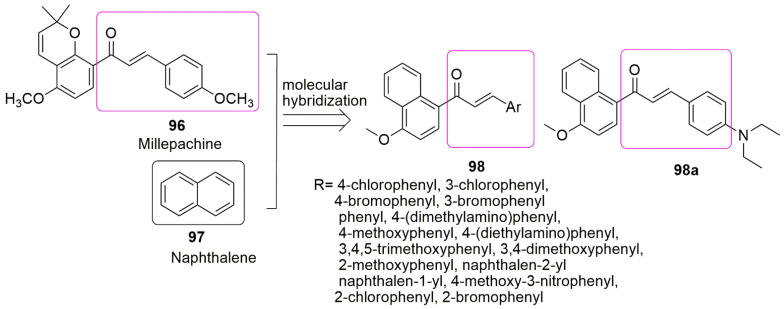
Molecular hybridization of chalcones and naphthalene moiety.

**Figure 29 ijms-23-04001-f029:**
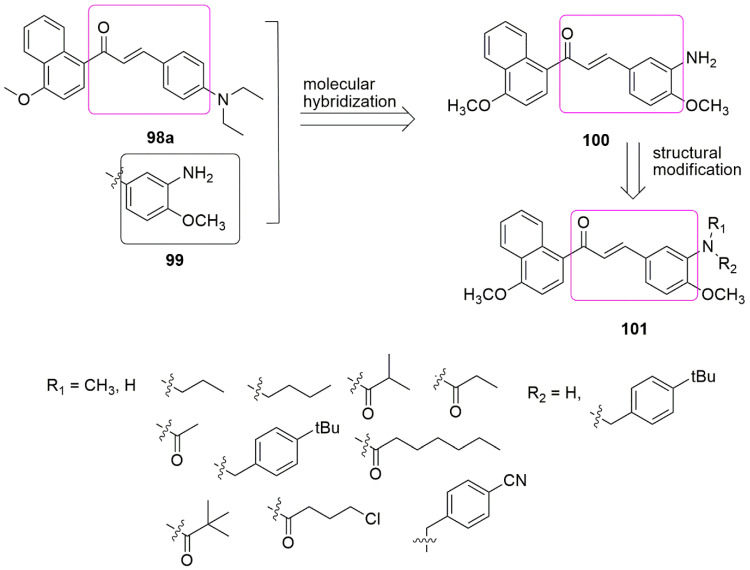
Chemical structures of chalcone hybrids evaluated against a panel of human cancer cell lines.

**Figure 30 ijms-23-04001-f030:**
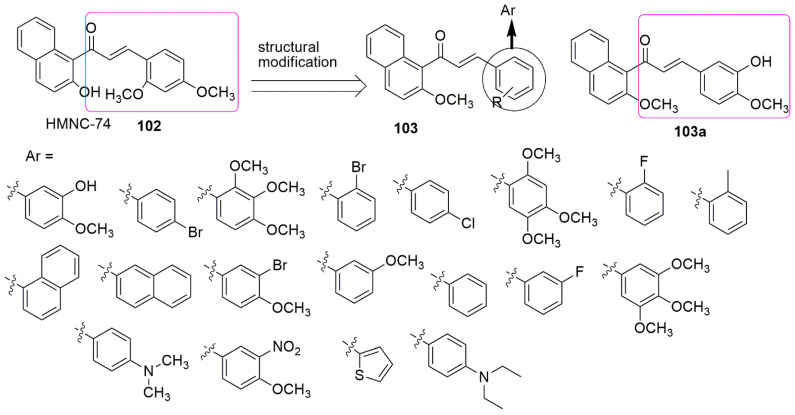
Chemical structures of naphthalene-chalcone hybrids.

**Figure 31 ijms-23-04001-f031:**
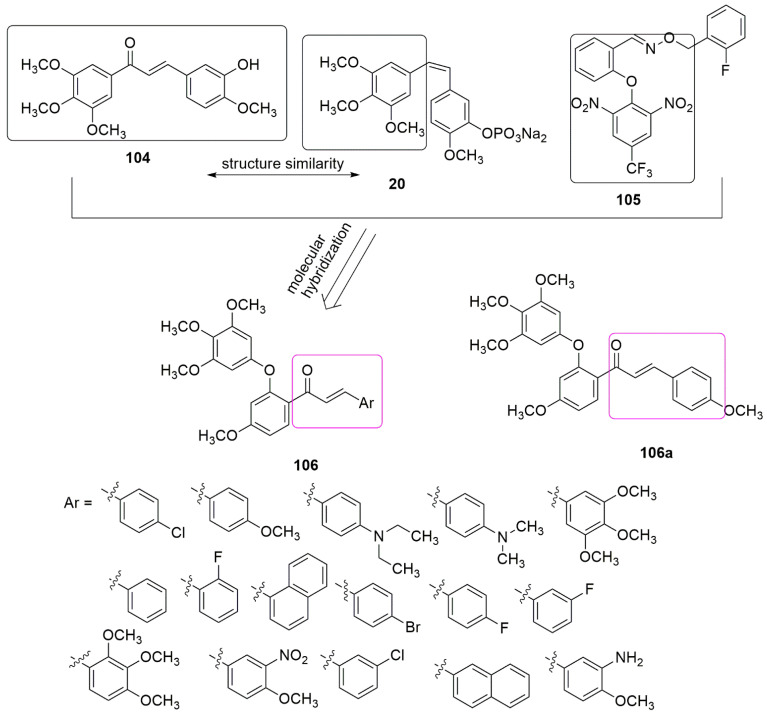
Chemical structures of tubulin polymerization inhibitor bearing chalcone, and diaryl ether hybrids.

**Figure 32 ijms-23-04001-f032:**
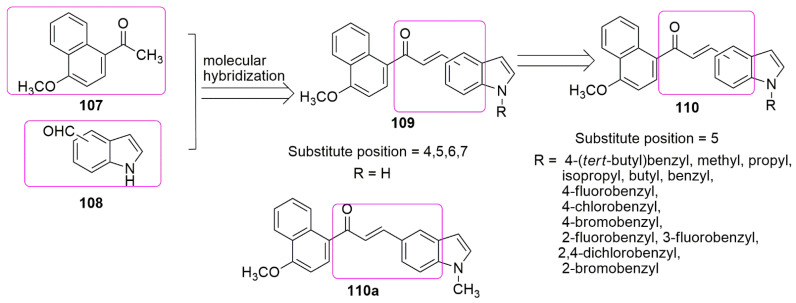
Chalcone derivatives containing indole and naphthalene moieties.

**Figure 33 ijms-23-04001-f033:**
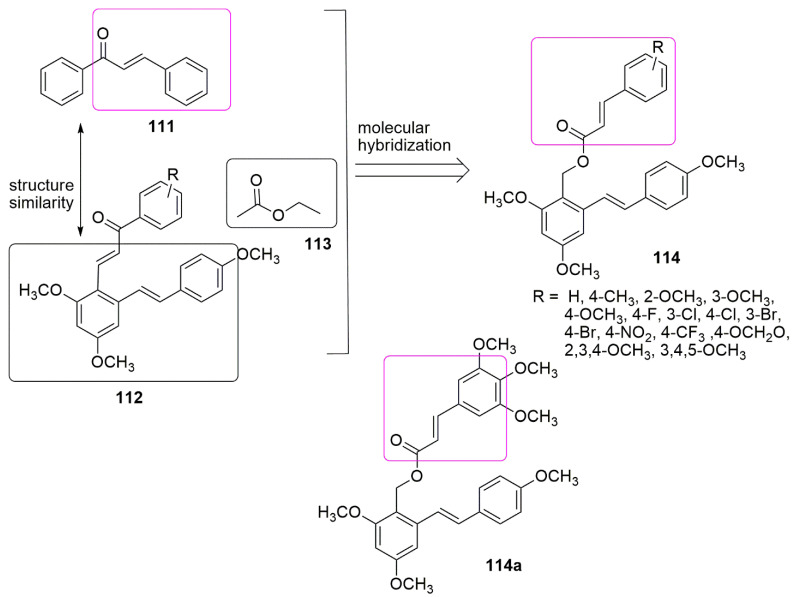
Chalcone derivatives bearing acyl ester functional group.

**Figure 34 ijms-23-04001-f034:**
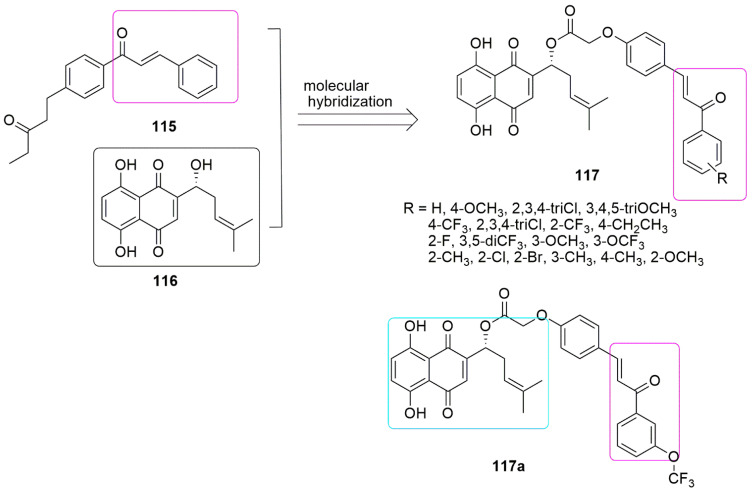
Chemical structures of novel chalcone-containing shikonin derivative.

**Figure 35 ijms-23-04001-f035:**
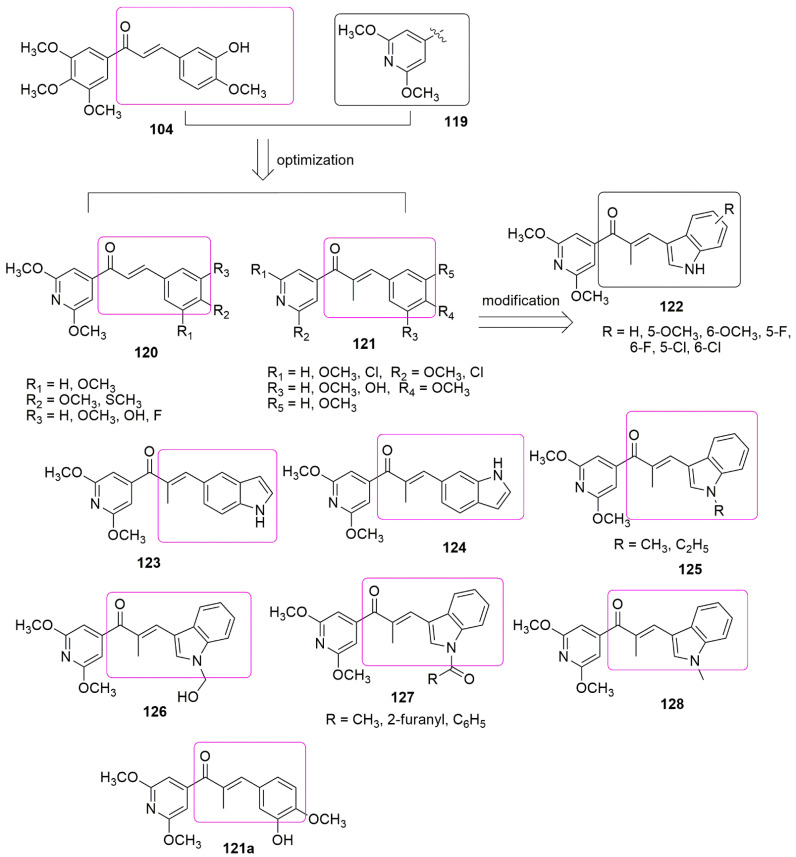
Chemical structures of pyridine-chalcone derivatives.

**Figure 36 ijms-23-04001-f036:**
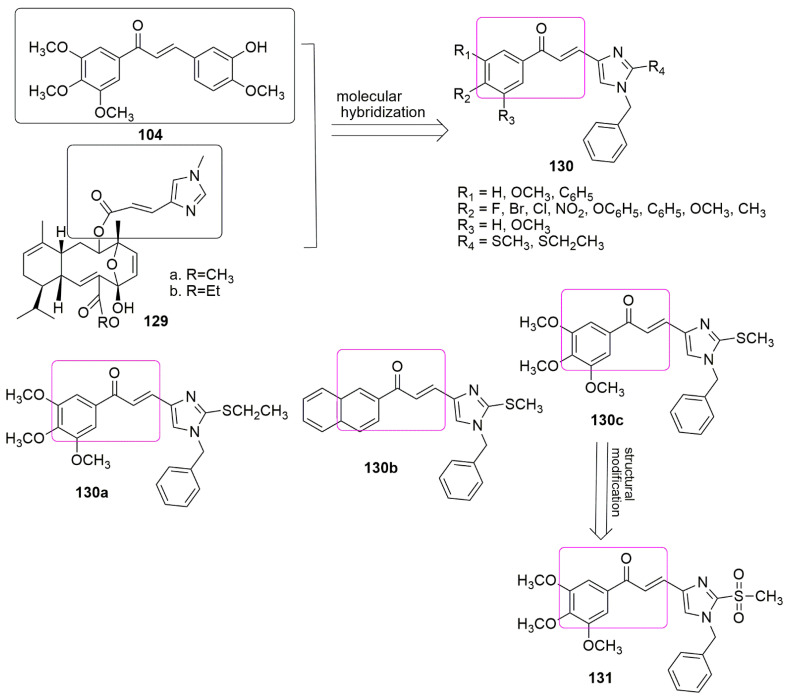
Chemical structures of imidazole–chalcone conjugates.

**Figure 37 ijms-23-04001-f037:**
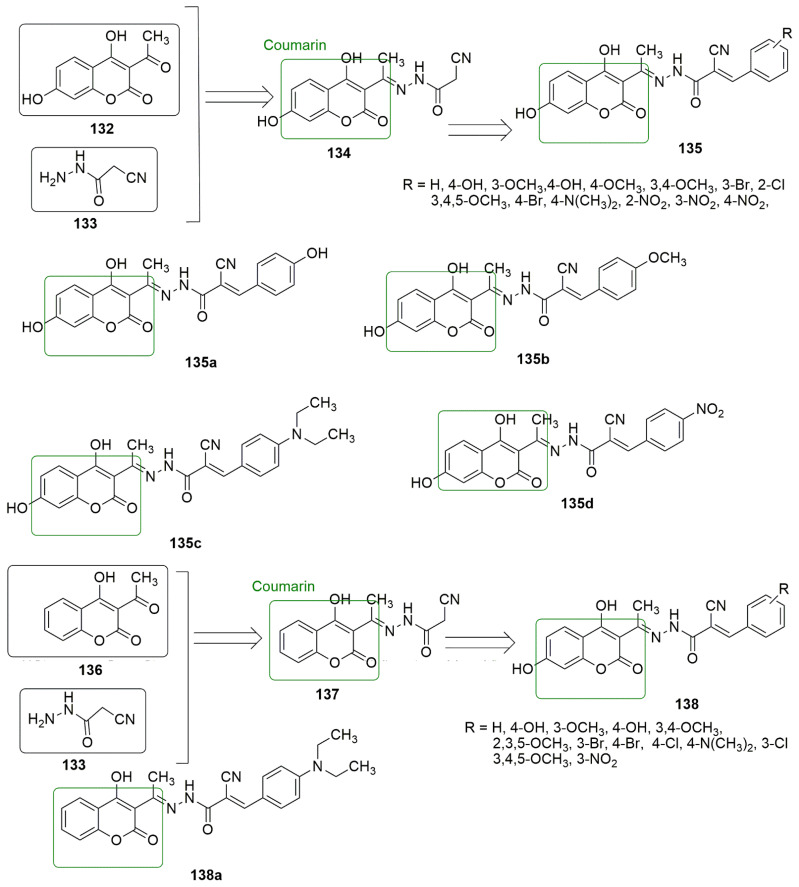
Novel coumarin-based acryloylcyanohydrazone compounds evaluated against a different panel of cancer cell lines.

**Figure 38 ijms-23-04001-f038:**
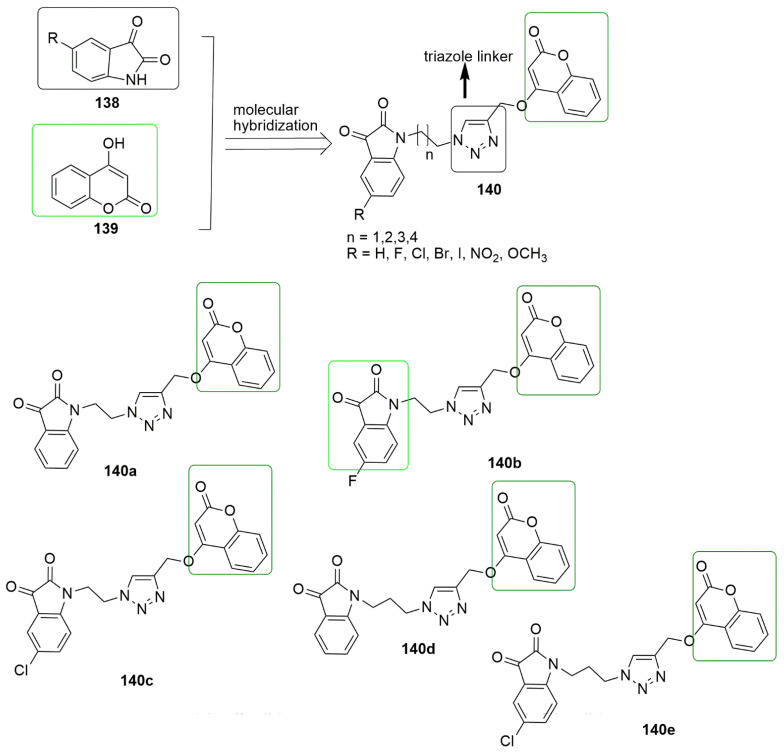
Chemical structures inhibitory values of triazole tethered isatin–coumarin bifunctional hybrids exhibiting anti-cancer and anti-cancer properties.

**Figure 39 ijms-23-04001-f039:**
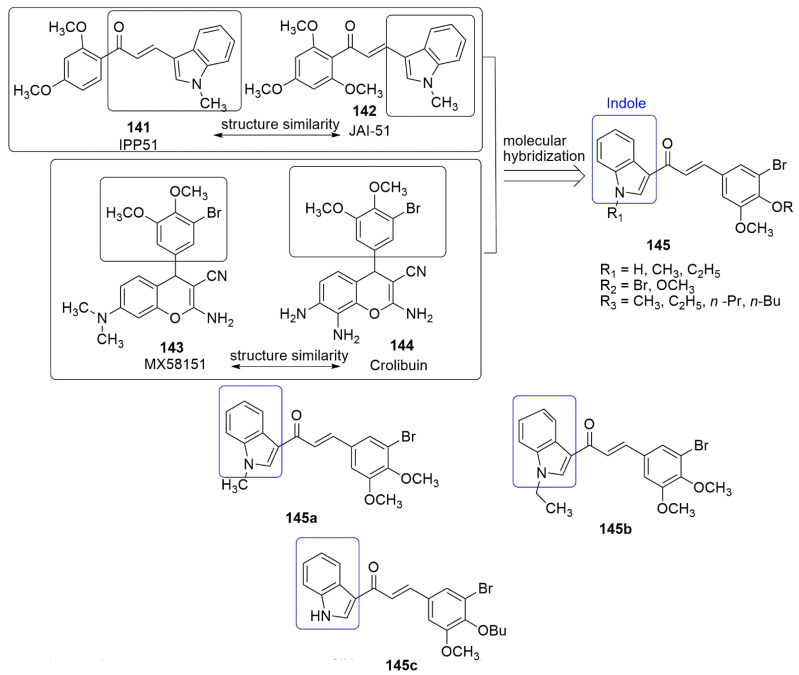
Amalgamation of indole–chalcone scaffolds as tubulin-targeting anti-cancer agent.

**Figure 40 ijms-23-04001-f040:**
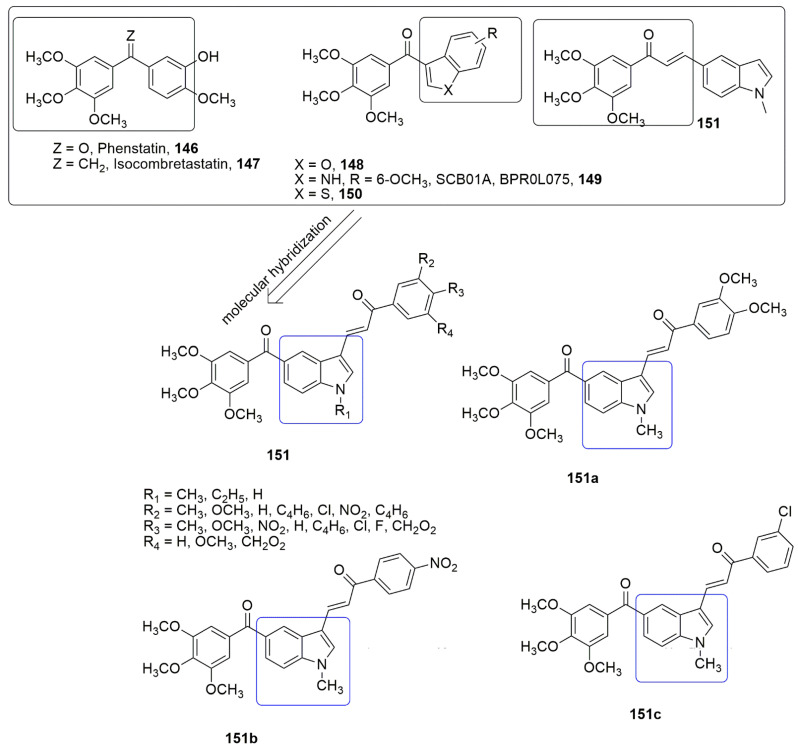
Novel phenstatin based indole-linked chalcone compounds.

**Figure 41 ijms-23-04001-f041:**
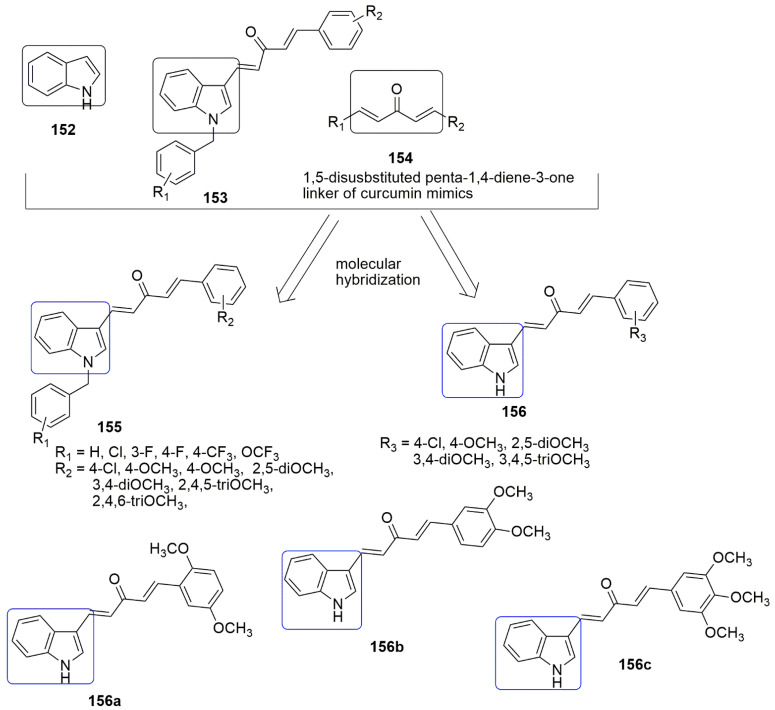
Chemical structure of curcumin-indole analogs evaluated against different panel cancer cell lines.

**Figure 42 ijms-23-04001-f042:**
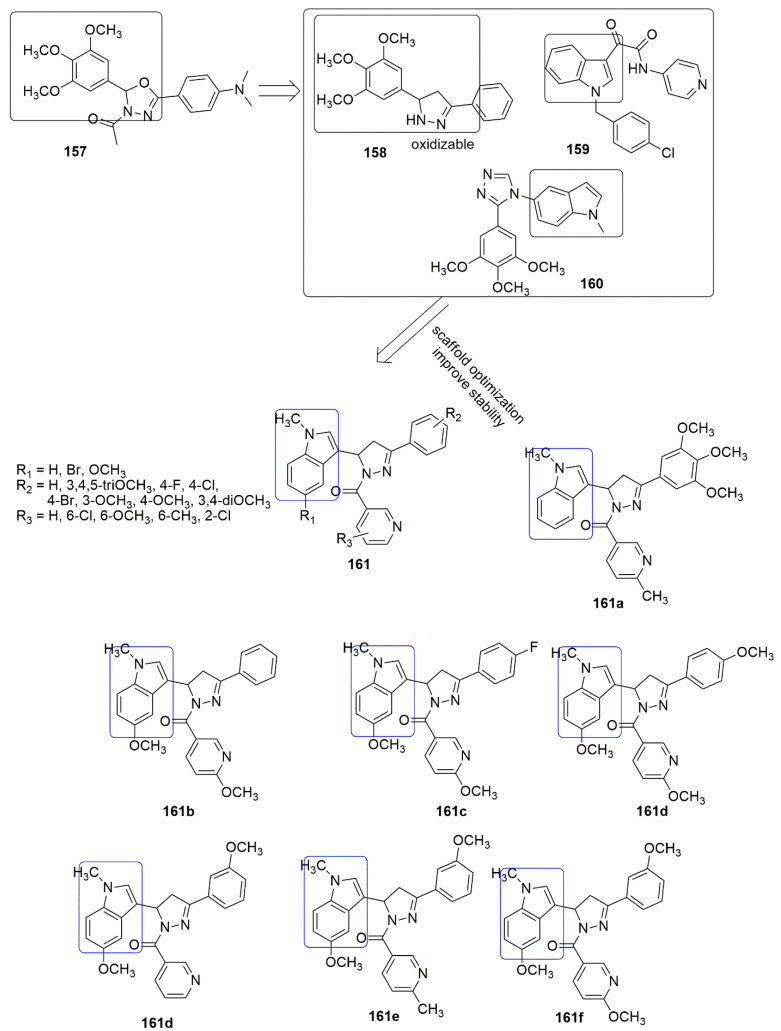
Nicotinoyl pyrazoline derivates bearing *N*-methyl indole exhibiting antiproliferative.

**Figure 43 ijms-23-04001-f043:**
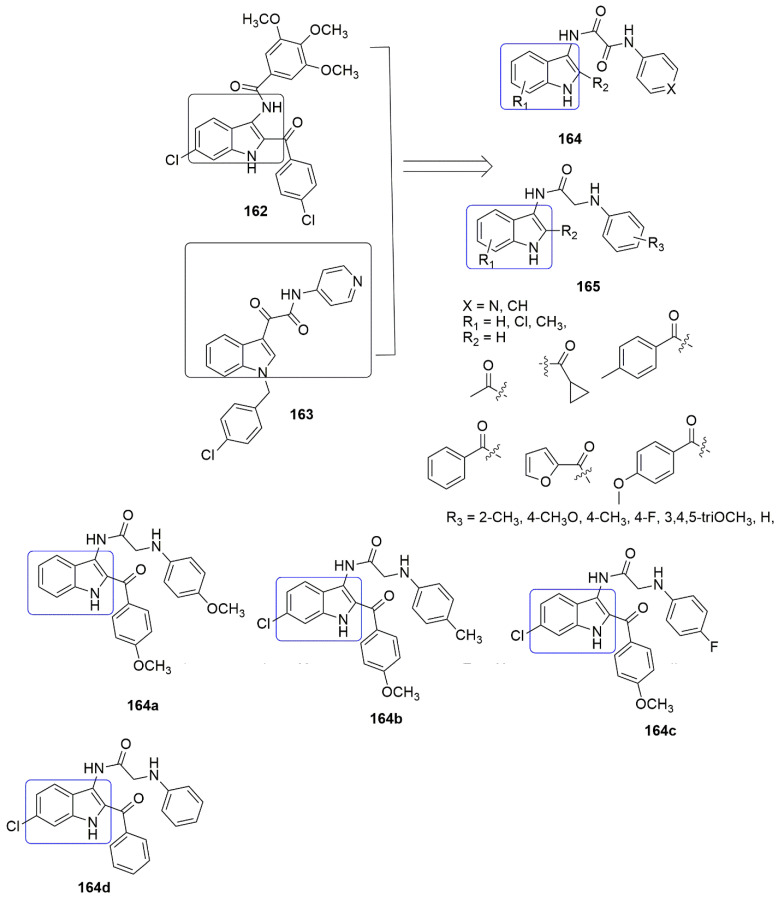
Novel indole-based oxalamide and aminoacetamide derivatives exhibiting anti-tubulin activity.

**Figure 44 ijms-23-04001-f044:**
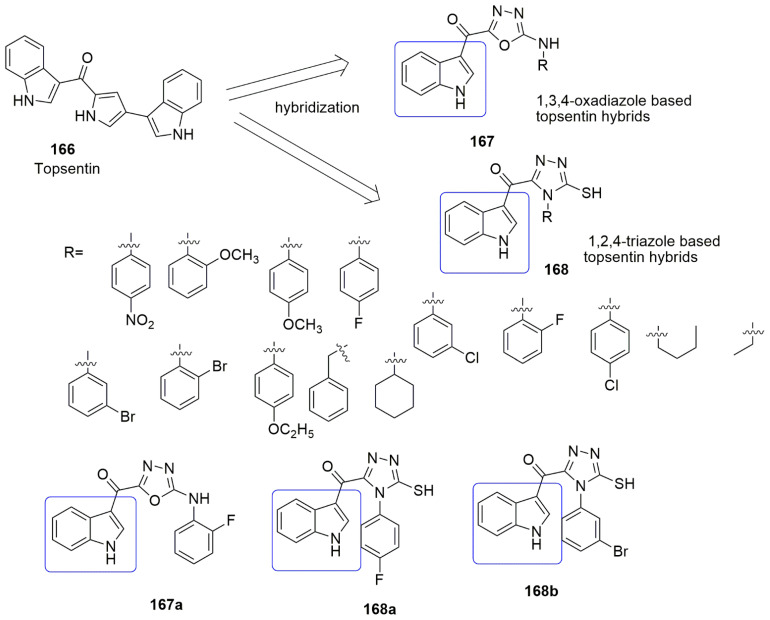
Chemical structure of indole-1,3,4-oxadiazole and indole-1,2,4-triazole-topsentin hybrids.

**Figure 45 ijms-23-04001-f045:**
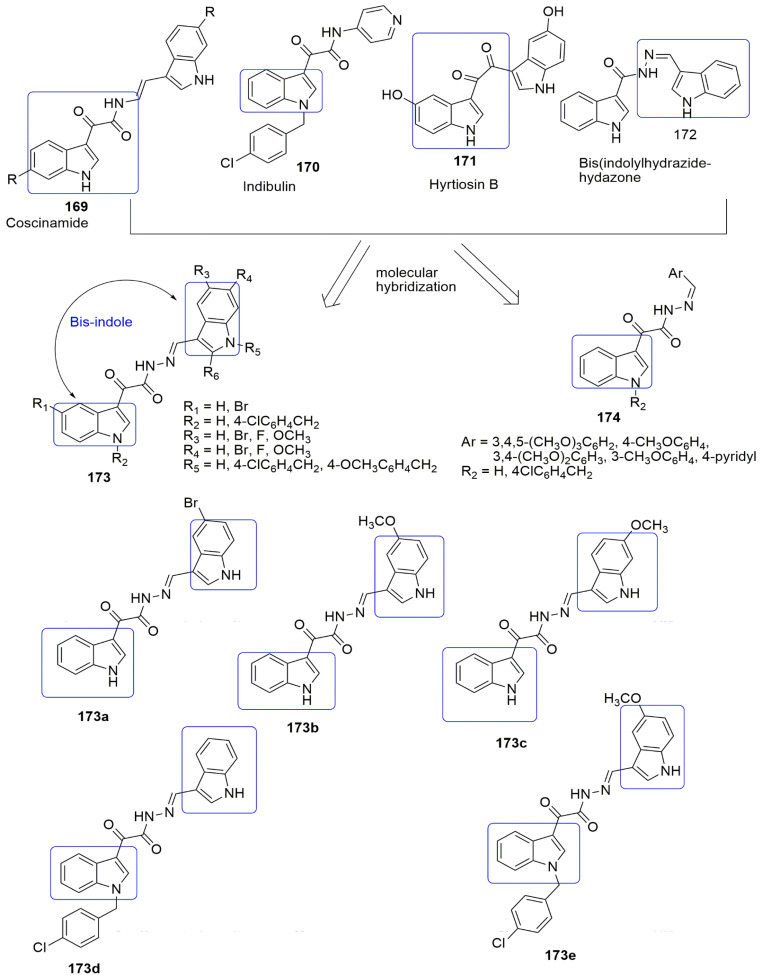
Chemical structures of bis(indolyl)ketohydrazide-hydrazones conjugates tested against different panel of cancer cell lines.

**Figure 46 ijms-23-04001-f046:**
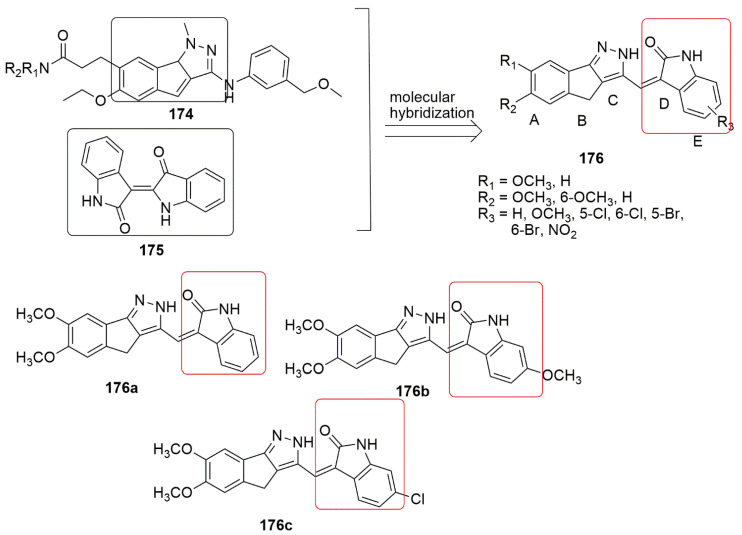
1, 4-dihydroindeno-[1,2-c] pyrazole in combination with oxindole.

**Figure 47 ijms-23-04001-f047:**
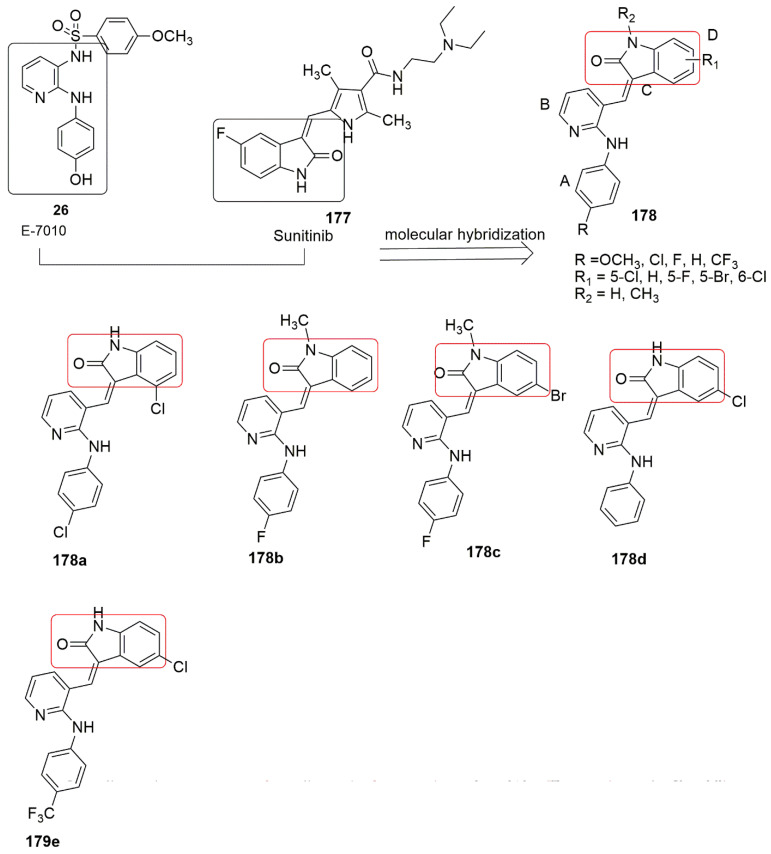
Chemical structures of anilinopyridyl-oxindole hybrids evaluated against a panel of four human cancer cell lines.

**Table 1 ijms-23-04001-t001:** Antiproliferative and anti-tubulin inhibition values of compounds **21a**–**e**, combretastatin A-4, and colchicine.

Compounds	GI50 ± SD (µM)				Tubulin Assembly
	A549	HeLa	HepG2	MCF-7	IC50 ± SD (µM)
**46a**	2.3 ± 0.27	4.5 ± 0.22	6.6 ± 0.26	11.7 ± 2.24	1.92 ± 0.24
**46b**	2.1 ± 0.61	3.4 ± 0.64	4.6 ± 0.33	7.9 ± 0.41	1.73 ± 0.52
**46c**	1.6 ± 1.17	2.7 ± 0.75	2.9 ± 1.08	4.3 ± 2.16	1.41 ± 3.67
**46d**	3.5 ± 0.28	6.6 ± 0.37	7.2 ± 0.22	13.5 ± 0.71	2.31 ± 2.23
**46e**	1.8 ± 0.61	2.9 ± 0.42	3.4 ± 0.11	5.2 ± 0.55	1.64 ± 4.76

**Table 2 ijms-23-04001-t002:** Antiproliferative and anti-tubulin activity of compound **48a**–**d**.

	IC50 ± SD (µM)				
Compounds	A549	HeLa	HepaG2	MCF-7	Tubulin Assembly
**48a**	0.20 ± 0.14	0.32 ± 0.08	0.41 ± 0.45	0.43 ± 0.73	1.82 ± 1.32
**48b**	0.17 ± 0.51	0.25 ± 1.22	0.37 ± 0.72	0.42 ± 0.06	1.63 ± 0.06
**48c**	0.15 ± 0.05	0.21 ± 0.67	0.33 ± 0.06	0.17 ± 0.23	1.52 ± 1.07
**48d**	0.27 ± 0.36	0.37 ± 0.05	0.46 ± 1.03	0.51 ± 0.74	1.97 ± 0.33
Colchicine	0.22 ± 0.12	0.36 ± 0.07	0.42 ± 0.12	0.44 ± 0.18	2.26 ± 0.25
CA-4	0.16 ± 0.04	0.24 ± 0.08	0.33 ± 0.12	0.18 ± 0.17	1.61 ± 0.31

**Table 3 ijms-23-04001-t003:** Antiproliferative activity of compounds **26a**–**c** and **27a**–**b**.

IC50 ± SD (µM)					
Compounds	A549	PC-3	HeLa	MDA-MB-231	L132
**58a**	5.42 ± 0.21	9.49 ± 1.21	10.14 ± 0.91	>25	ND
**58b**	6.11 ± 2.18	8.13 ± 2.71	>25	13.53 ± 2.72	ND
**58c**	22.41 ± 3.84	1.46 ± 1.89	8.57 ± 1.54	>25	ND
**59a**	12.44 ± 1.56	11.19 ± 0.67	21.34 ± 1.57	>25	ND
**59b**	2.21 ± 0.12	3.15 ± 0.41	7.29 ± 0.67	5.71 ± 1.14	69.25 ± 5.95
Nocodazole	2.39 ± 0.14	1.96 ± 0.24	3.48 ± 0.52	2.13 ± 0.25	ND

**Table 4 ijms-23-04001-t004:** Antiproliferative activity of compounds **28a** and **28b** measured in IC50 values (µM).

Compound	hPERT RPE-1	Capan-1	HCT-116	NCI-H460	DND-41	HL-60	K-562	MM.1 S	Z-138
**62a**	4.3	0.3	0.6	0.4	0.2	0.3	2.1	1.5	0.4
**62b**	1.7	0.2	0.4	0.6	0.3	0.2	1.4	1.3	0.4
Docetaxel	0.0553	0.0088	0.0017	0.0024	0.0125	0.0072	0.0152	0.0118	0.0142

**Table 5 ijms-23-04001-t005:** Antiproliferative activity of compound **83a**–**c** measured in *IC*_50_ values (µM).

Compounds	A-2780	A-2780/RCIS	MCF-7	MCF/MX	HUVEC
**81a**	0.50 ± 0.07	1.21 ± 0.22	1.11 ± 0.26	1.66 ± 0.27	9.94 ± 2.68
**81b**	1.44 ± 0.16	1.13 ± 0.22	1.79 ± 0.26	1.67 ± 0.17	7.82 ± 2.44
**81c**	0.38 ± 0.08	0.85 ± 0.11	2.45 ± 0.65	2.45 ± 0.53	5.02 ± 1.34

**Table 6 ijms-23-04001-t006:** In vitro cell growth inhibitory effects of compounds GI_50_ (nM) and inhibition of tubulin assembly *IC*_50_ (nM).

Compounds	HeLa	A549	HT-29	MDA-MB-231	RS4;4	Jurkat	Kasumi-1	Tubulin Assembly
**88a**	0.5 ± 0.03	4.2 ± 0.7	1.9 ± 0.7	17.1 ± 9.4	0.5 ± 0.05	0.7 ± 0.002	0.4 ± 0.1	0.99 ± 0.07
**88b**	34 ± 0.35	45 ± 0.94	4.7 ± 0.5	591 ± 35	7 ± 0.8	3 ± 0.02	15 ± 2	1.1 ± 0.1
**88c**	4.2 ± 0.3	1.1 ± 0.3	0.6 ± 0.04	123 ± 8.8	0.2 ± 0.05	0.1 ± 0.002	0.3 ± 0.09	0.84 ± 0.05
CA-4	4.0 ± 1.0	180 ± 50	3100 ± 100	n.d	0.8 ± 0.2	5.0 ± 0.6	ND	0.64 ± 0.01

**Table 7 ijms-23-04001-t007:** The in vitro antiproliferative activities *IC*_50_ (µM) of compounds **130a**–**130c**.

Compouds	A-549	MCF-7	HEPG 2	MCF-7 MX
**130a**	11.7 ± 1.62	14.85 ± 2.73	63.43 ± 5.33	14.94 ± 1.91
**130b**	66.47 ± 4.85	>100	>100	54.86 ± 2.94
**130c**	7.05 ± 1.12	9.88 ± 1.56	21.97 ± 2.23	20.2 ± 2.76
**131**	ND	18.33 ± 3.15	>100	>100
CA-4	0.86 ± 0.23	0.43 ± 0.14	0.63 ± 0.17 1	49 ± 0.47

ND—not determined.

**Table 8 ijms-23-04001-t008:** IC50 (µM) values of active hybrids against sensitive human cancer cell lines and tubulin polymerization.

Compounds	THP-1 (Leukemia)	COLO-205 (Colon)	HCT-116 (Colon)	Tubulin Polymerization
**140a**	0.73	3.45	3.04	1.06
**140b**	1.99	6.67	5.41	3.55
**140c**	5.47	8.87	5.77	-
**140d**	5.03	5.71	5.18	6.32
**140e**	6.99	10.66	9.66	-

**Table 9 ijms-23-04001-t009:** IC50 (µM) values of active hybrids (**161a**–**161g**) against sensitive cancer cell lines and tubulin polymerization.

Compound	GI50 ± SD (µM)		CC50 ± SD (µM)		IC50 ± SD (µM)	
	MCF-7	A549	HepG2	HeLa	293 T	Tubulin Assembly
**161a**	0.18 ± 0.01	0.23 ± 0.02	0.31 ± 0.02	0.21 ± 0.02	>300	2.3 ± 0.2
**161b**	0.99 ± 0.01	1.11 ± 0.04	1.33 ± 0.05	1.1 ± 0.05	>300	2.9 ± 0.1
**161c**	0.48 ± 0.01	0.71 ± 0.03	0.22 ± 0.02	0.41 ± 0.02	>300	2.6 ± 0.2
**161d**	0.56 ± 0.03	0.45 ± 0.01	0.23 ± 0.02	0.18 ± 0.01	>300	2.3 ± 0.1
**161e**	0.81 ± 0.04	1.02 ± 0.05	0.91 ± 0.01	1.27 ± 0.05	125.40 ± 1.82	2.1 ± 0.2
**161f**	0.09 ± 0.01	0.59 ± 0.02	0.029 ± 0.01	0.034 ± 0.01	>300	1.6 ± 0.1
**161g**	0.13 ± 0.01	0.47 ± 0.03	0.19 ± 0.01	0.34 ± 0.02	>300	1.9 ± 0.1
CA-4	0.14 ± 0.02	0.31 ± 0.02	0.17 ± 0.02	0.092 ± 0.01	>300	2.1 ± 0.2

## Data Availability

Not applicable.

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
