# Peer review of "A Review of the Recent Developments of Molecular Hybrids Targeting Tubulin Polymerization"

_ijms, 2022, doi:10.3390/ijms23074001_

Round 1

Reviewer 1 Report

I have reviewed the article submitted by Jack Tusynski and co-authors describing literature examples of molecular hybrids targeting tubulin polymerization.

This review article, which cites 130 literature references, plans to describe the different strategies that authors reported for the design and development of molecular hybrids, targeting the polymerization of tubulin into microtubules.

This review article is quite interesting and, regarding to the potential interest for the scientific community, and especially for researchers working on tubulin polymerization inhibitors, could be published in IJMS.

Nonetheless, there are too many imprecisions (see below the remarks concerning “molecular hybrids”), some errors in figures’ numbering, formulas errors. Consequently, I strongly suggest to write a corrected version of this interesting review project and send back to the editorial office for a complete new evaluation.

Please find below a non-exhaustive list of corrections that must be done before a new evaluation of this paper.

In the abstract, G2-M should be written G2/M.

When used for the first time in the article, abbreviations must be detailed (except those that are well known, such as DNA…): eg, NMSC…

Authors have to pay attention that they must use the official abbreviations for journals, rather than entire names, as defined by the internet web site CASSI Search Tool.

Please rename “Figure A” (text) / “Figure 13 (??) (figure caption page 2) by Figure 1. Modify this sentence “according to their mortality rate” as this figure seems to indicate their global distribution (and not the mortality rate).

I do not agree with the last classification “(iv) natural products” as a natural product could also be (ii) microtubule-targeting agents: please modify it.

I would better write “anti-cancer drug action” than “cancer drug action.”

Figure numbering in the text is wrong: alphabetical letters in the text, whereas numbers in the figure captions. This must be corrected (Figure B -> Figure 2…)

In Figure 2, include a taxane derivative rather the three vinblastine/vincristine/vinorelbine compounds (that act similarly…).

Please correct this sentence: “medicinal chemistry compounds” eg by “compounds originated from different medicinal chemistry strategies”.

Authors MUST check ALL the chemical structures depicted in this paper.

Indeed, there is an error in the chemical structure of combretastatin A-4 as the (E)-isomer is drawn (wrong stereoisomer…) whereas the (Z)-isomer is the most active compound, contained in the natural source…

Second error for bromoindirubin where the bromine atom is not at the right position.

In the same figure, please draw the stereochemistry (dashed bond) of the methyl esters in vinblastine/vincristine. These two structures should have been drawn in the same orientation.

Figure 2 caption is missing (and what is H3N3?), and I guess that the text was pasted within the figure caption…

And my feeling is that a part of the text is missing… Indeed, the transition between the introducing part and the triazoles derivatives is kind of puzzling… No other paragraph indication than “Introduction” lies in the text… So I guess that part of the text is missing to mention the architecture of the manuscript and the philosophy of hybridization…

So, please provide corrections that will explain the content of the manuscript and how the hybrids will be classified. Actually, this is maybe the “Achille heel” of this review article as nothing describes the structural modifications and hydridization strategies. This would clearly help the reader for a better understanding, but the hybridization concept is not clearly explained in the manuscript.

Page 4: please mention the organ type the cell lines (SGC-7901, A549, and HT-1080) originate.

Furthermore, I really do not understand the figure numbering… Page 5 and page 6, “new” figures 1 and 2 are described??? This MUST be corrected before considering a new evaluation of this article.

I see that, as they explained in the introduction, authors wish to present the SAR for all the depicted molecules, which is of real interest. Nonetheless, the presentation should include in the figures the activity of the molecules, especially with a variation of R,R1, R2…groups. For example, in figure 1 page 5, biological activity should be briefly indicated for the different compounds.

All the molecules must be clearly identified. As an example, in “Figure 1, page 5”, compound 3 vary depending on R/R1/R2 groups but we do not have any idea of the exact molecules correspond to this general formula and which number are described in the cited reference.

In this figure (and in all the figures of this article…), some R groups (here, R3) are missing, and authors should not use plain arrows to indicate the heterocyclic motif. Concerning this last point, the drawings must be homogeneous: sometimes the heterocycle name is written using a colour corresponding to the square that surrounds this heterocyclic structure and sometimes not, sometimes the square box line are dashed and sometimes plain, sometimes circles are there and sometimes squares…

The description of the biological activities must be also written clearly: Tubulin Polymerization Inhibition and cellular anti-proliferative effects. Authors have also to mention at which concentration the compounds have shown G2/M phase arrest.

Tables must be separated from figures, with a separated title/caption: eg pages 26, 27, 37, 41 where the tables are “included” in the figure…

To conclude with my reviewing, I think that authors must modify the organization of the article to make it clearer (especially for the hybridization concept/strategies), appropriately describe the different series and correct mistakes. Particularly, the presentation of the “hybrids” remains to my opinion not clear at all: this article deals only with the structural motifs within TPI chemical structures and does not present a real concept of molecular hybrids.

I quote the abstract of one article (Current Medicinal Chemistry 2007, 14, 1829-1852): “Molecular hybridization is a new concept in drug design and development based on the combination of pharmacophoric moieties of different bioactive substances to produce a new hybrid compound with improved affinity and efficacy, when compared to the parent drugs.”. Authors must clarify this concept, through all the examples they would like to present. With some more explanations, this would give an interesting review article.

In the conclusion (and in the text), authors must be more precise in defining the term of “hybrid molecules”, which is not only related with the “mixing” of two chemical structures but must also mention the biological activity of each molecular scaffold.

Author Response

Attached please see our detailed response to the comments made by Reviewer #1

Reviewer 2 Report

This article reviews publications of the last five years on new molecular hybrid compounds that inhibit microtubules. The article summarizes in particular the chemical structures of different families of hybrid compounds, and their respective pharmacological potencies. The article has an encyclopedic character, but it is difficult to read for non-specialists without a profound knowledge in organic chemistry (and, sorry, it is dry and reads a bit like a telephone directory…).

Overall, I think this article is poorly structured: the introduction ends abruptly, and without any separating headline, the presentation of diverse compounds starts. Also, there is not much discussion that puts the reviewed data in a larger context.

Several sections of the manuscript must be revised: in particular, the numbering and the labelling of the figures is largely incorrect. Specific examples:

  • Page 2: the first figure is labelled “Figure 13” in the legend, and “Figure A” in the text, line 36; it should be corrected to read “Figure 1”.
  • Page 3: “Figure 2” in the legend is mentioned as “Figure B” in the text, line 62.
  • Page 3: the legend to Figure 2 makes no sense. Is this meant to be the legend for the third figure (also to be revised in line 112)?
  • Page 4, line 132: the reference to Figure 2 makes no sense.
  • Page 5, line 134: “Figure 1” should be “Figure 3”, and all following figures must be re-numbered correctly, both in the legends and in the main text.

Minor points:

Line 54: “microtubules … contributing to the formation of the mitotic spindle…” – Saying that microtubules “contribute” to the formation of spindles is a bit of an understatement, since microtubules represent their main structural components.

Line 57: “… cell cycle arrest in G2/M phase…” - G2 and M are two distinct phases, and inhibition of microtubules leads to the specific arrest in M phase (mitosis).

Line 65: “dynamics of tubulin” should rather be changed to “microtubule dynamics”.

Author Response

Attached please find our detailed replies to the comments made by Reviewer #2.
